# Immune subset-committed proliferating cells populate the human foetal intestine throughout the second trimester of gestation

Nannan Guo [1], Na Li[1,2], Li Jia[1], Qinyue Jiang[1], Mette Schreurs [1], Vincent van Unen [1,3], Susana M. Chuva de Sousa Lopes [4], Alexandra A. Vloemans[1], Jeroen Eggermont [5], Boudewijn Lelieveldt[5], Frank J. T. Staal [1], Noel F. C. C. de Miranda [6], M. Fernanda Pascutti[1,7] ✉ & Frits Koning [1,7] ✉

The intestine represents the largest immune compartment in the human body, yet its development and organisation during human foetal development is largely unknown. Here we show the immune subset composition of this organ during development, by longitudinal spectral flow cytometry analysis of human foetal intestinal samples between 14 and 22 weeks of gestation. At 14 weeks, the foetal intestine is mainly populated by myeloid cells and three distinct CD3⁻CD7⁺ ILC, followed by rapid appearance of adaptive CD4⁺, CD8⁺ T and B cell subsets. Imaging mass cytometry identifies lymphoid follicles from week 16 onwards in a villus-like structure covered by epithelium and confirms the presence of Ki-67⁺ cells in situ within all CD3⁻CD7⁺ ILC, T, B and myeloid cell subsets. Foetal intestinal lymphoid subsets are capable of spontaneous proliferation in vitro. IL-7 mRNA is detected within both the lamina propria and the epithelium and IL-7 enhances proliferation of several subsets in vitro. Overall, these observations demonstrate the presence of immune subset-committed cells capable of local proliferation in the developing human foetal intestine, likely contributing to the development and growth of organized immune structures throughout most of the 2nd trimester, which might influence microbial colonization upon birth.

The developing human fetus generates tolerogenic and protective immune responses, in preparation for antigen exposure during pregnancy and after birth[1]. Multiple immune cell types develop and mature at different gestational stages, and diverse immune cells seed lymphoid and peripheral organs, including lymph nodes, skin, intestine, kidney and lung[1-3]. The intestine represents the largest compartment of the immune system. Immunohistochemistry (IHC) analysis of human fetal intestinal samples has shown that T cells are detectable in the lamina

[1]Department of Immunology, Leiden University Medical Center, Leiden, Netherlands. [2]State Key Laboratory of Zoonotic Diseases, Institute of Zoonoses, College of Veterinary Medicine, Jilin University, Changchun, China. [3]Institute for Immunity, Transplantation and Infection, Stanford University, Stanford, CA, USA. [4]Department of Anatomy, Leiden University Medical Center, Leiden, Netherlands. [5]Department of Radiology, Leiden University Medical Center, Leiden, Netherlands. [6]Department of Pathology, Leiden University Medical Center, Leiden, Netherlands. [7]These authors contributed equally: M. Fernanda Pascutti, Frits Koning. ✉e-mail: M.F.Pascutti@lumc.nl; F.Koning@lumc.nl

propria and epithelium from 12–14 weeks of gestation and T cells increase in abundance afterwards[4]. Moreover, evidence for co-localization of T and B cells has been presented[5,6]. Nevertheless, the development and complexity of the immune compartment during human fetal life is understudied for reasons of the scarcity of material and ethical concerns.

Lymphoid tissues are important locations for the initiation of adaptive immune responses[7], and their formation requires interaction between lymphoid-tissue inducer cells (LTi) and stromal cells[8–10]. B cells also contribute to the maturation of gut-associated lymphoid tissues (GALT), including Peyer's patches (PP) and isolated lymphoid follicles (ILFs)[11]. In the intestinal mucosa, antigen-presenting cells (APC) and in particular dendritic cells (DC) are located throughout the intestinal lamina propria, and can recruit naive T cells to the different lymphoid tissues[12,13]. In addition, memory CD4+ T cells were frequently observed in the human fetal intestine[14], and fetal intestinal TNF-α+CD4+ T cells promote mucosal development[15,16].

Improvements in high-dimensional imaging and flow cytometry techniques provide new opportunities to map the niche and development of the human fetal intestinal immune compartment with higher resolution than ever before. Here we used multi-parametric spectral flow cytometry to determine the composition of the fetal immune system throughout a major part of the 2nd trimester. Furthermore, we have combined this with an imaging mass cytometry (IMC)-based approach to gain detailed insight into the spatial localization of the fetal intestinal immune system in the tissue context in time[17,18].

The results indicate dynamic changes in the composition of the fetal intestinal immune system, the continuous presence of proliferation-associated Ki-67+cells in all detected immune subsets and the early formation of lymphoid follicles (LF) harboring B cells, as well as various types of innate lymphoid cells (ILC), T cells and myeloid cells. In addition, Ki-67+ cells were frequently observed in such LFs. Moreover, we observed that subsets of CD3−CD7+ ILCs, CD4−CD8α−, CD4+ and CD8α+ T cells displayed a shared phenotype characterized by the expression of CD69, CD117, CD161 and CCR6 and that these cells were preferentially found in the LFs. Together these results point very early formation of complex lymphoid structures in the human fetal intestine and suggest that local proliferation of immune subset-committed cells contributes to the development and growth of organized immune structures throughout most of 2nd trimester.

## Results

### Identification of Ki-67-positive cells within all immune subsets in the developing human fetal intestine

To investigate the development of the human fetal intestinal immune compartment in time, we applied a 26-antibody flow cytometry panel to single-cell suspensions of intestinal samples from gestational weeks 14 to 22 (antibodies listed in Supplementary Table 1). We pooled the spectral flow cytometer-acquired data on CD45+ immune cells of all samples (2.6 million cells) in a single analysis with optSNE.

Based on marker expression profiles (Fig. 1a), seven major immune cell populations were identified, corresponding to CD3+CD4+ T cells, CD4+CD25+CD127−FoxP3+ T regulatory cells (Treg), CD3+CD8+ T cells, CD3+CD4−CD8− double-negative (DN) T cells, CD20+HLA-DR+ B cells, CD3−CD7+ ILCs and CD11c+ myeloid cells (Fig. 1b, Supplementary Figs. 1, 2). As we have reported previously[14], we observed a dominant presence of CD4+FOXP3− conventional T cells in the fetal intestine, approximately 50% of which expressed CD161 and CD45RO while lacking CCR7 and CD45RA, indicative of a CD161+ effector memory T cell type (T_EM) (Fig. 1a). All CCR7−CD45RA− T_EM cells were CD127+ and CD69+, and differential expression of CD117 and CCR6 was observed (Fig. 1a), reflecting additional phenotypic diversity. The remainder of the CD4+ conventional T cell population expressed CCR7, implying a

CD45RA+ naïve T cell (T_N) or central CD45RO+ memory T cell (T_CM) phenotype (Fig. 1a).

Moreover, three clusters of CD8+ T cells (a. CCR7+CD8α+CD8β+, b. CCR7−CD8α+CD8β+ and c. CD8α+CD8β−), and a single DN T cell cluster were observed (Fig. 1a, b). Also, the differential expression of CD8α, CD25, CD45RA, CD117, CD127, RORγt and CCR6 revealed the presence of three distinct subpopulations of CD3−CD7+ ILCs, one of which expressed CD117, CD127 and RORγt, compatible with LTi cells (Fig. 1a, b). Two B cell clusters were present, one of which was HLA-DR+CD1c+CD45RA+CCR6+, while another was HLA-DR^low CD1c−CD45RA^low CCR6− (Fig. 1a, b). Finally, a single cluster of myeloid cells was observed, but differential expression of CD11c, CD163, HLA-DR, and CD1c points towards heterogeneity within this myeloid subset (Fig. 1a, b). Strikingly, we observed clusters of Ki-67-positive cells within all identified cell subsets, indicative of cell proliferation (Fig. 1b, c). Similar results were obtained in two additional independent experiments (Supplementary Figs. 1, 2).

Together, these results indicated substantial heterogeneity in the developing fetal intestinal immune compartment and revealed Ki-67-positive cells in all adaptive and innate cell subsets detected.

### Stable presence of Ki-67 positive cells throughout gestational week 14 to 22

To obtain further information on the development of the fetal intestinal immune system and the presence of Ki-67 positive cells in time, we analyzed the tSNE-plots of the individual human fetal samples from gestational weeks 14 to 22 (Fig. 2a, cluster partition as shown in Fig. 1a, Supplementary Fig. 3a, b). The percentage of CD45+ cells acquired from each sample is summarized in Fig. 2b. At week 14, the intestinal immune compartment was primarily composed of one B cell subset, CD3−CD7+ ILC subsets and myeloid cells, followed by the rapid emergence of the other immune subsets in the weeks thereafter (Fig. 2a). The frequencies of myeloid cells and CD3−CD7+ ILC steadily decreased in time (Fig. 2c, d), while the percentage of B cells, CD4+ T cells, and Tregs increased in time (Fig. 2e–g), and CD8+ and DN T cells remained more constant (Fig. 2h, i). From week 15 onwards, CD4+ T cells were the dominant immune lineage in the intestine samples (Fig. 2f).

For comparison, we also analyzed unaffected intestinal samples from pediatric and adult individuals (Fig. 3). Here we observed differences in immune composition between the fetal and *ex utero* samples, exemplified by an increase in the frequency of CD8+ T cells and a decrease in the frequency of CD4+ T cells and CD3−CD7+ ILCs in the pediatric and adult samples in comparison to the fetal samples (Fig. 3a–e). Moreover, in the fetal samples we identified Ki-67+ cells within all detected immune subsets at all time points (Fig. 2j, Supplementary Fig. 3c–k), where the actual percentage of Ki-67+ cells within the CD45+ cells remained stable in time (Fig. 2k). In contrast, the frequency of Ki-67+ immune cells was substantially lower in the pediatric and adult samples compared to fetal samples (Fig. 3f–h). In addition, the percentages of CD127+ ILCs and CD127+ CD8+, CD4+ and DN T cells was higher in the fetal samples compared to the *ex utero* samples (Fig. 3i).

Thus, the composition of the fetal intestinal immune system is distinct from that in pediatric and adult samples. While at week 14, CD3−CD7+ ILC, B cells and myeloid cells are abundant in the fetal intestine, from week 15 onwards, adaptive T and B cell populations dominate the immune compartment. Moreover, stable clusters of Ki-67+ cells are present in all immune lineages throughout a major part of the second trimester.

### Molecular analysis confirms peripheral expansion of intestinal T cells

T cell receptor excision circles (TREC) provide a molecular means to quantitively assess the replication history of T lymphocytes. Different

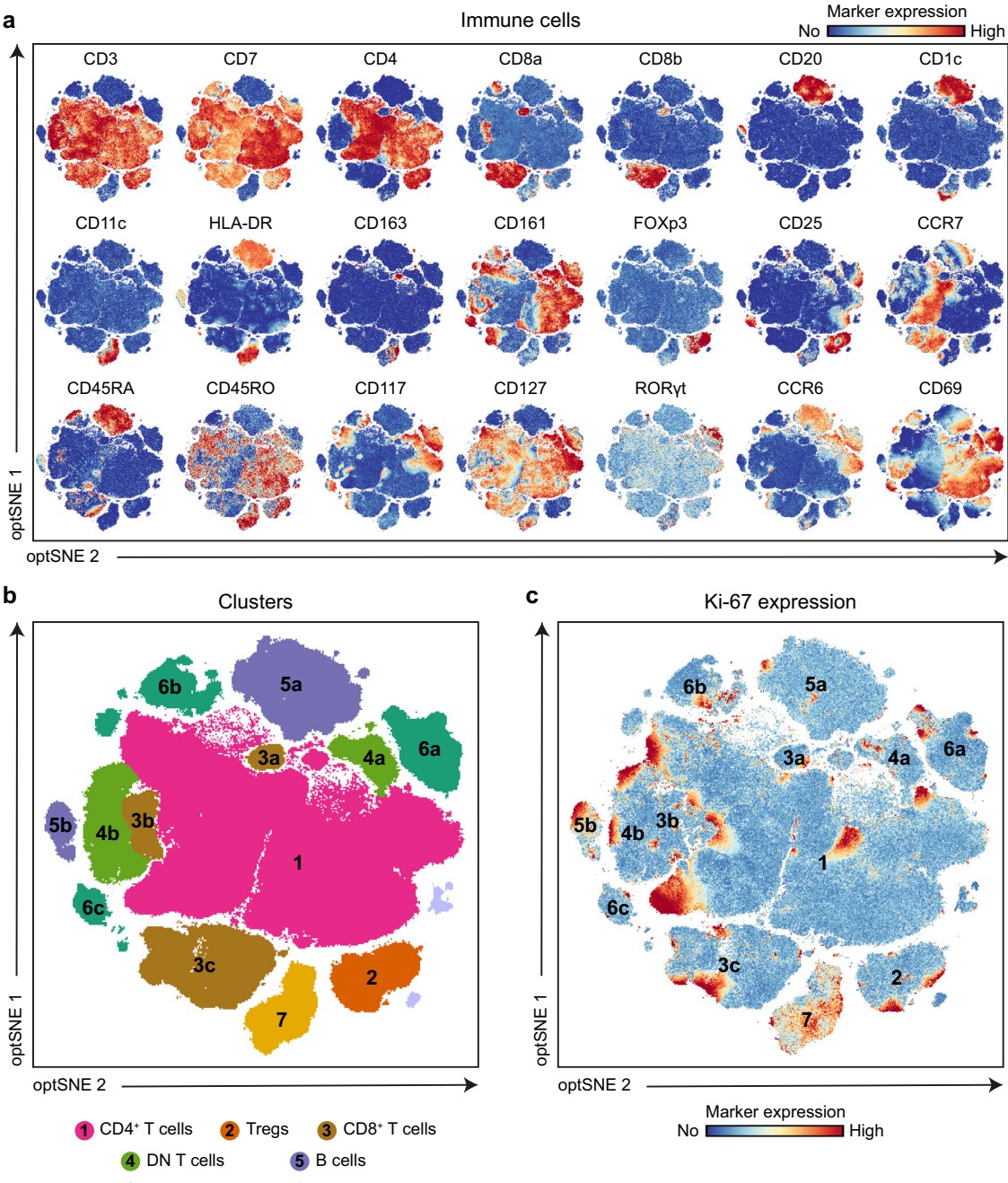

**Fig. 1 | Identification of Ki-67⁺ cells within all immune subsets in the human fetal intestine. a** A collective optSNE was performed on CD45⁺ immune cells from 12 human fetal intestinal samples. Each dot represents a single cell. In total, 2.7 × 10⁶ CD45⁺ immune cells were analyzed by OMIQ. Colors represent relative expression of indicated immune markers. **b** Based on the relative marker expression profile, each immune lineage was identified and color-coded. **c** Display of the Ki-67 expression within the CD45⁺ immune cells. Colors represent relative expression of Ki-67. Data represent three independent experiments.

TREC assays exist, which allow the measurement of extensive proliferation in the thymus (Vγ-Jγ TREC) or mostly peripheral expansion after thymic egress (δREC-ψJα TREC). We applied the δREC-ψJα TREC to fetal intestinal samples, because of its greater sensitivity to measure post-thymic proliferation. Similar to earlier results in purified cord blood naive T cells, the T cells in the fetal intestine had undergone 3-4 cell divisions at week 20 and this number increased to 6.5 divisions at week 22 (Fig. 4a).

In addition, we re-analyzed a previously generated single cell RNAseq dataset on purified fetal intestinal CD4⁺ T cells[14]. tSNE analysis revealed a distinct cluster of CD4⁺ T cells expressing *MKI67*, *CCNB2* and *CDK1*, in line with active proliferation (Fig. 4b). In addition, these cells co-expressed *IL7R* and *CD69*, indicative of responsiveness to stimulation with IL-7 and tissue residency, respectively. Integrated analysis of the cell cycle gene expression patterns placed the *MKI67* positive cells in the G2M phase (Fig. 4c), indicative of active proliferation.

Finally, we analyzed the expression of genes associated with T cell receptor rearrangements. Expression of *RAG1* and *LIG4* were only detected in few cells, while no expression of *RAG2* and *DNTT* was found (Fig. 4d), arguing against T cell receptor rearrangements taking place in the fetal intestinal compartment.

Altogether, these results point to recent or active proliferation in the intestinal T cell compartment.

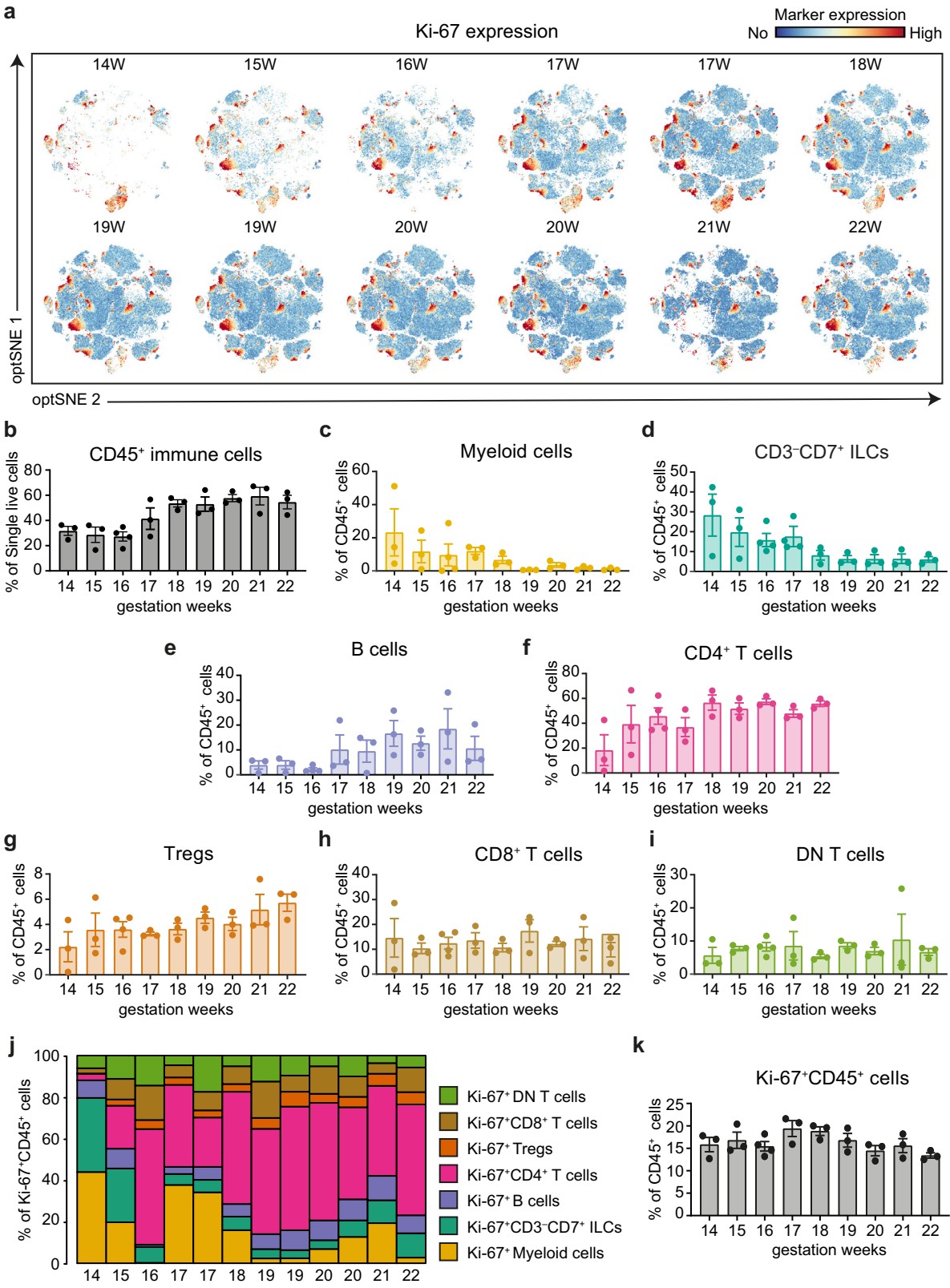

**Fig. 2 | Presence of Ki-67⁺ cells throughout gestational week 14 to 22. a** Display of the Ki-67 expression in the optSNE plots of the individual fetal intestinal samples from week 14 through 22. Colors represent the relative expression of Ki-67. Data are representative of three independent experiments. **b** The frequency of CD45⁺ immune cells within all human fetal intestine samples analyzed ($n = 28$). Error bars indicate mean ± s.e.m. **c–i** Overview of the frequency of each immune lineage within the CD45⁺ immune cells of all human fetal intestine samples analyzed ($n = 28$). Error bars indicate mean ± s.e.m. **j** Overview of the distribution of Ki-67⁺ cells in the indicated immune lineages from gestational week 14 through 22, the results shown are from the samples shown in panel **a**. **k** The percentage of Ki-67⁺ cells within the CD45⁺ immune cells of all human fetal intestine samples analyzed ($n = 28$). Error bars indicate mean ± s.e.m.

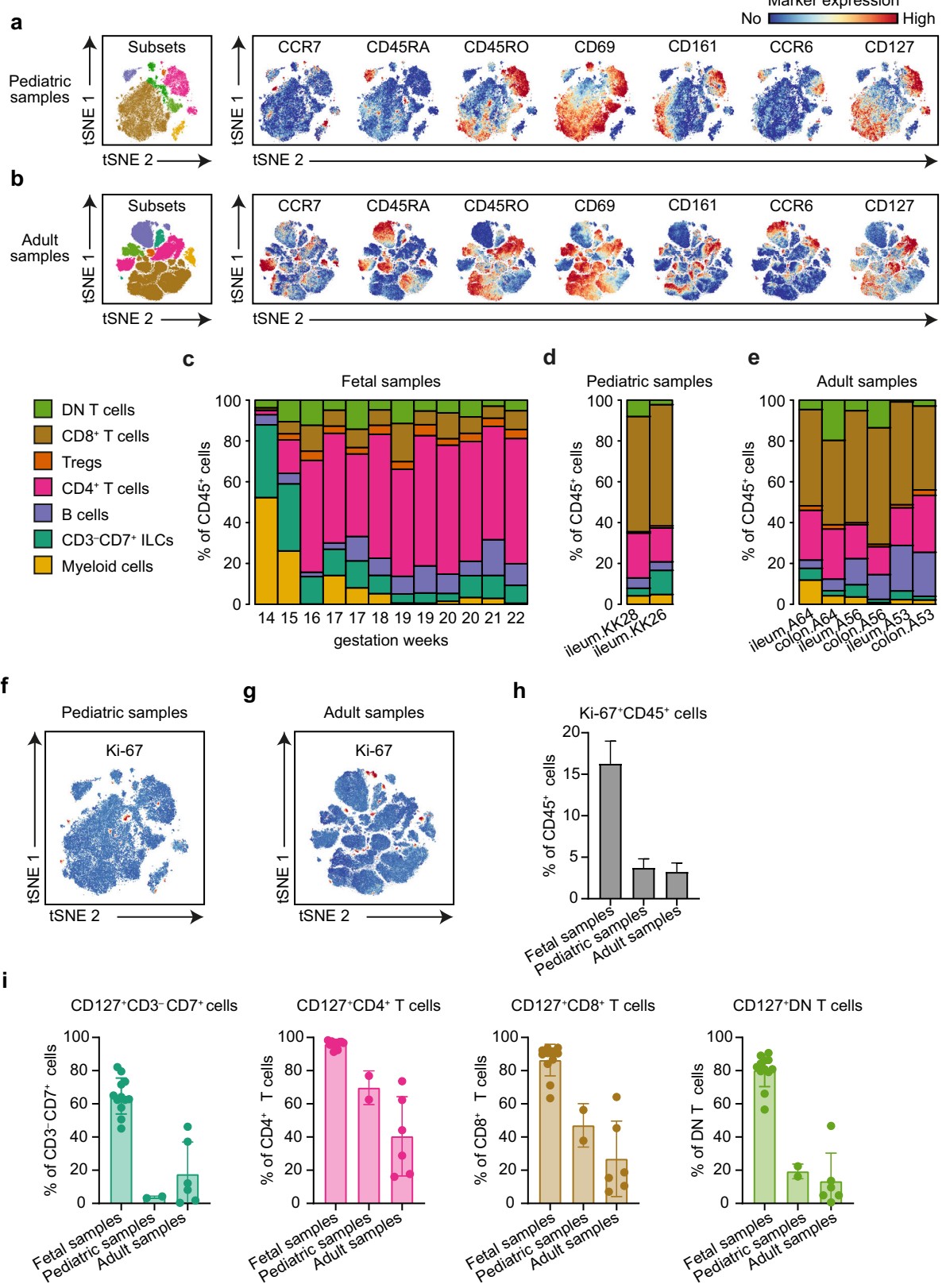

**Fig. 3 | Identification of immune cells composition of pediatric and adult samples. a, b** tSNE plots of the data from (**a**) pediatric (*n* = 2) and (**b**) adult (*n* = 6) samples, where the major immune subsets are indicated and the expression of selected markers is shown. **c–e** Percentages of the major immune subsets in the samples analyzed. **f–h** tSNE plots displaying Ki-67 expression in samples from (**f**) pediatric and (**h**) adult samples. Error bars indicate mean ± s.e.m. **g** Percentage of Ki-67+ cells within CD45+ immune cells of all samples analyzed. **i** Frequency of CD127/IL-7Rα expressing cells within CD3−CD7+ ILCs, CD4−CD8− DN T cells, CD4+ T cells and CD8+ T cells in fetal, pediatric and adult samples. Each dot represents an individual sample. Error bars indicate mean ± s.e.m.

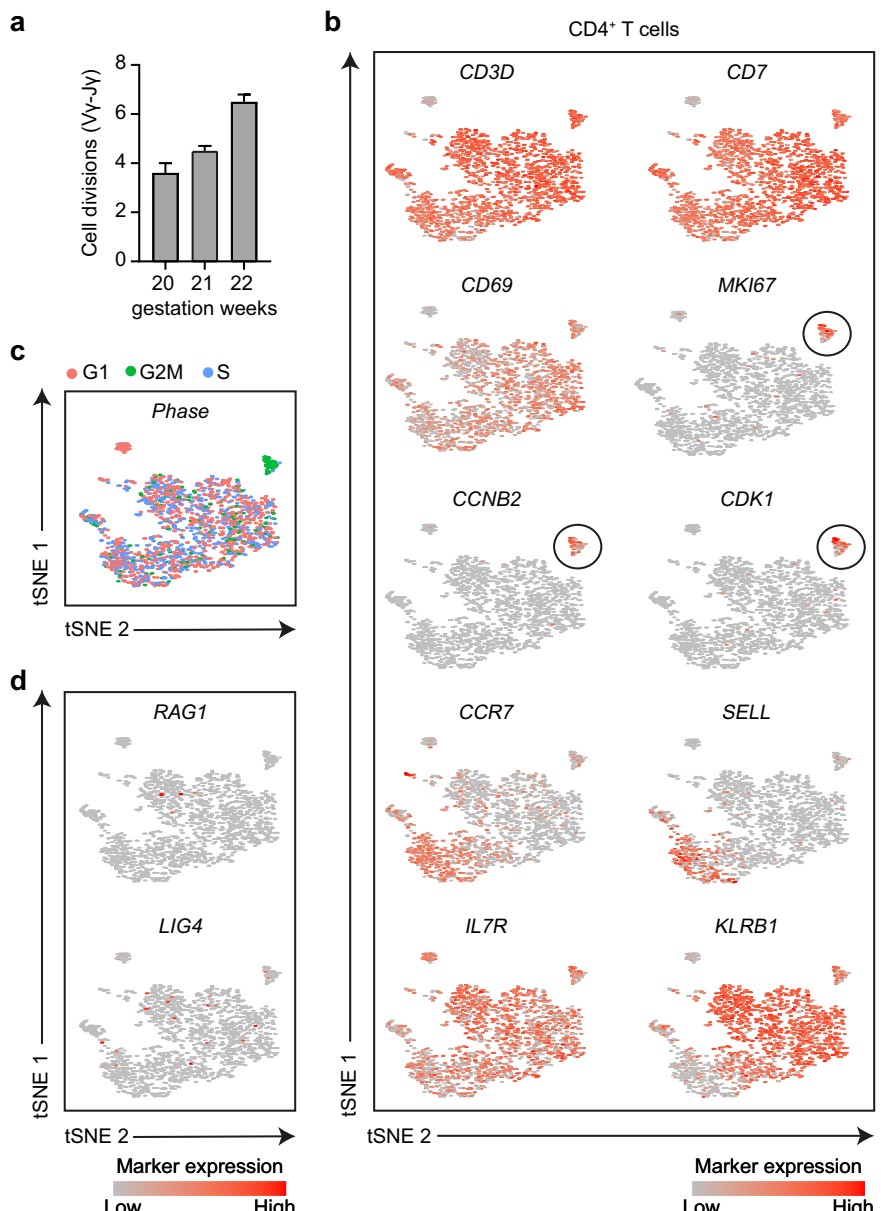

**Fig. 4 | T cell receptor excisions circle (TREC) and single cell RNA-seq analysis confirms peripheral expansion of intestinal T cells. a** Proliferative history in human fetal intestinal samples (*n* = 3). Error bars indicate mean ± SD. **b–d** t-SNE embedding showing 1804 CD4⁺ T cells from a human fetal intestine analyzed by single-cell RNA-sequencing. The log-transformed expression levels of indicated immune markers (**b**), the phase of the cell-division cycle (**c**) and genes involved in TCR rearrangement (**d**) are shown. Each dot represents a single cell. The proliferative CD4⁺ T cells are encircled (**b**).

## Lymphoid Follicles form early in the development of the human fetal intestine

To explore the development and spatial organization of the fetal intestinal immune system in situ, we employed imaging mass cytometry (IMC) on tissue sections of human fetal intestinal samples from different gestational ages (Fig. 5). For this we developed and optimized antibody panels that incorporated markers to distinguish tissue structure, and innate and adaptive immune subsets (Supplementary Table 2–4). Based on E-cadherin and CD45 expression patterns, the epithelium and immune cells could be readily identified (Fig. 5a). Most CD45⁺ immune cells were localized in the lamina propria in all fetal intestine samples (Fig. 5a). Strikingly, aggregates of immune cells were present in all samples in villus-like structures covered by epithelium, termed lymphoid follicles (LF) hereafter, which presented with a higher density of CD45⁺ immune cells in time (Fig. 5a, encircled by boxes). Analysis of the expression of the immune lineage markers CD3,

CD7, CD20, HLA-DR and CD163 revealed the presence of T cells, CD3⁻CD7⁺ ILCs, B cells and two subsets of myeloid cells within the LFs (Fig. 5b–f). At week 16 of gestation, HLA-DR⁺ myeloid cells were the primary immune cell population in the LFs, while CD3⁻CD7⁺ ILCs, B cells and a few T cells were also detected. At later time points in gestation, increased numbers of B and T cells were detected in these structures.

We next aimed to obtain a comprehensive overview of the organization of all detected immune subsets in a single image from a 21-week sample where a large LF was visible. For this we made use of "*Cytosplore Imaging*", which was adapted from *Cytosplore*⁺ᴴˢᴺᴱ [19], to allow analysis of the IMC data at the single-pixel level. Supplementary Fig. 4 provides an exploration workflow of "*Cytosplore Imaging*", indicating how the tissue-specific pixels are extracted from the original IMC dataset, where pixels representing a single cell type are assigned a color and projected back onto the original image. The approach allows

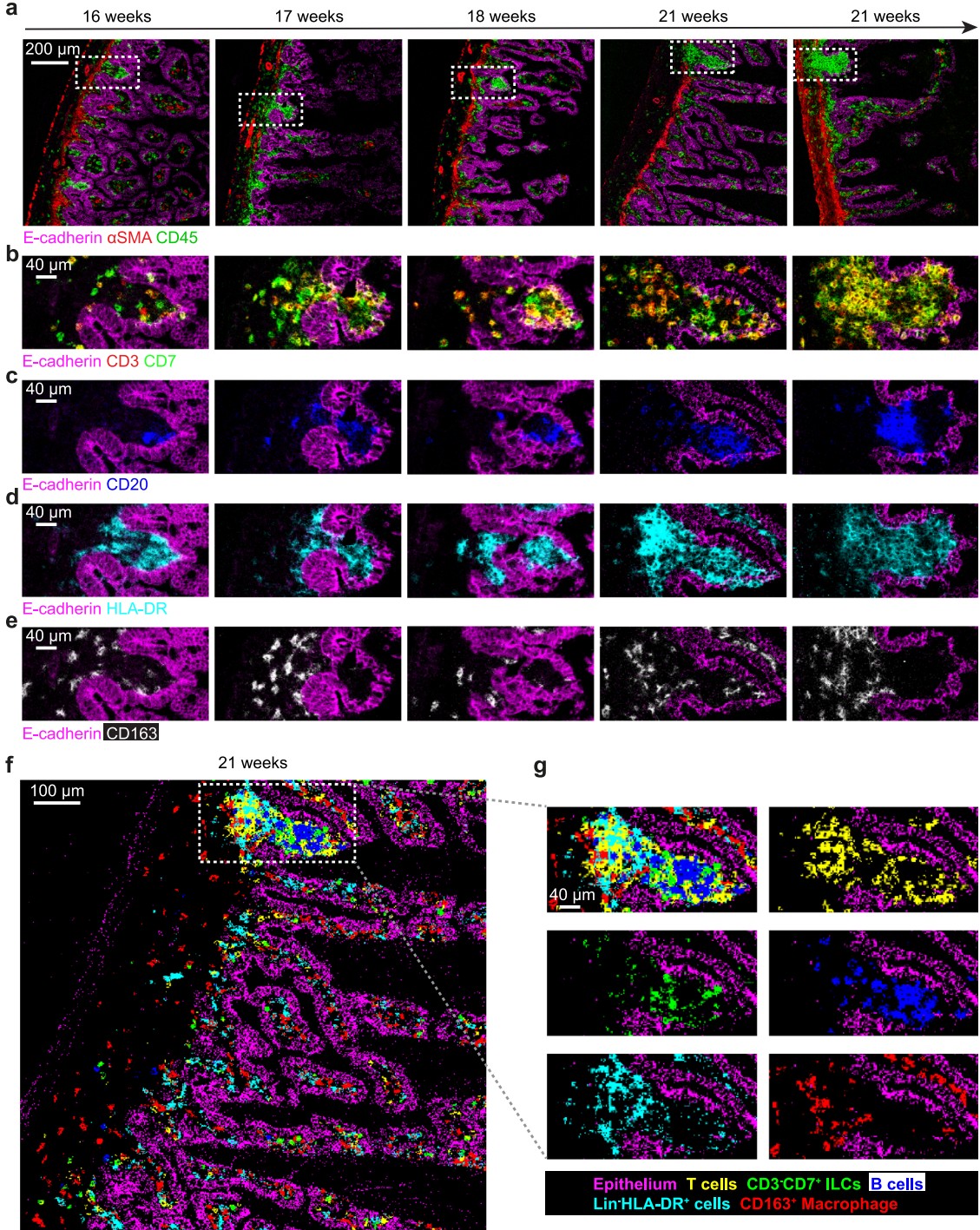

**Fig. 5 | Identification and visualization of the immune cell composition of LFs in human fetal intestinal samples by imaging mass cytometry. a** Visualization of E-cadherin, αSMA, and the immune cell marker CD45 reveals the overall structure of the fetal intestinal samples from week 16 through 21 and the location of LF (indicated by boxes) within those samples. Scale bar, 200 μm. Identification of CD3⁺CD7⁺ T cells (yellow) and CD3⁻CD7⁺ ILCs (green) by the overlay of CD3 (red) and CD7 (green) (**b**); CD20⁺ B cells (blue) (**c**); Lin⁻HLA-DR⁺ myeloid cells (magenta) (**d**); CD163⁺ macrophages (white) (**e**). Scale bar, 40 μm. **f** Visualization of the location of the immune subsets in a representative human fetal intestine sample from gestational week 21 by pixel analysis in "*Cytosplore imaging*". Scale bar, 100 μm. The image is representative of three independent experiments. **g** Overview of the individual immune cell populations in the LF by pixel analysis. Scale bar, 40 μm.

the simultaneous visualization of up to 9 cell types with unique colors in a single image at 1 μm pixel resolution (Supplementary Fig. 4a–d). We applied this approach to visualize the composition and organization of immune cells in the 21-week LF. For this, we embedded the CD45⁺/dim pixels at the data level and projected the immune subset-related pixels back onto the image. The resulting image revealed the distribution of T cells (CD3⁺CD7⁺), CD3⁻CD7⁺ ILCs

(CD3⁻CD7⁺), B cells (CD20⁺HLA-DR⁺), Lin⁻HLA-DR⁺ myeloid cells (CD3⁻CD7⁻CD20⁻CD163⁻HLA-DR⁺) and CD163⁺ macrophages (CD163⁺HLA-DR⁺) in situ in a single region (Fig. 5f, Supplementary Fig. 4e). The overview of the image indicates the presence of a B cell follicle in close contact with CD3⁻CD7⁺ ILCs and T cells in the top of the villus-like structure, while underneath a cluster of HLA-DR⁺ and CD163⁺ myeloid cells with T cells was present. Moreover, the HLA-DR⁺ myeloid

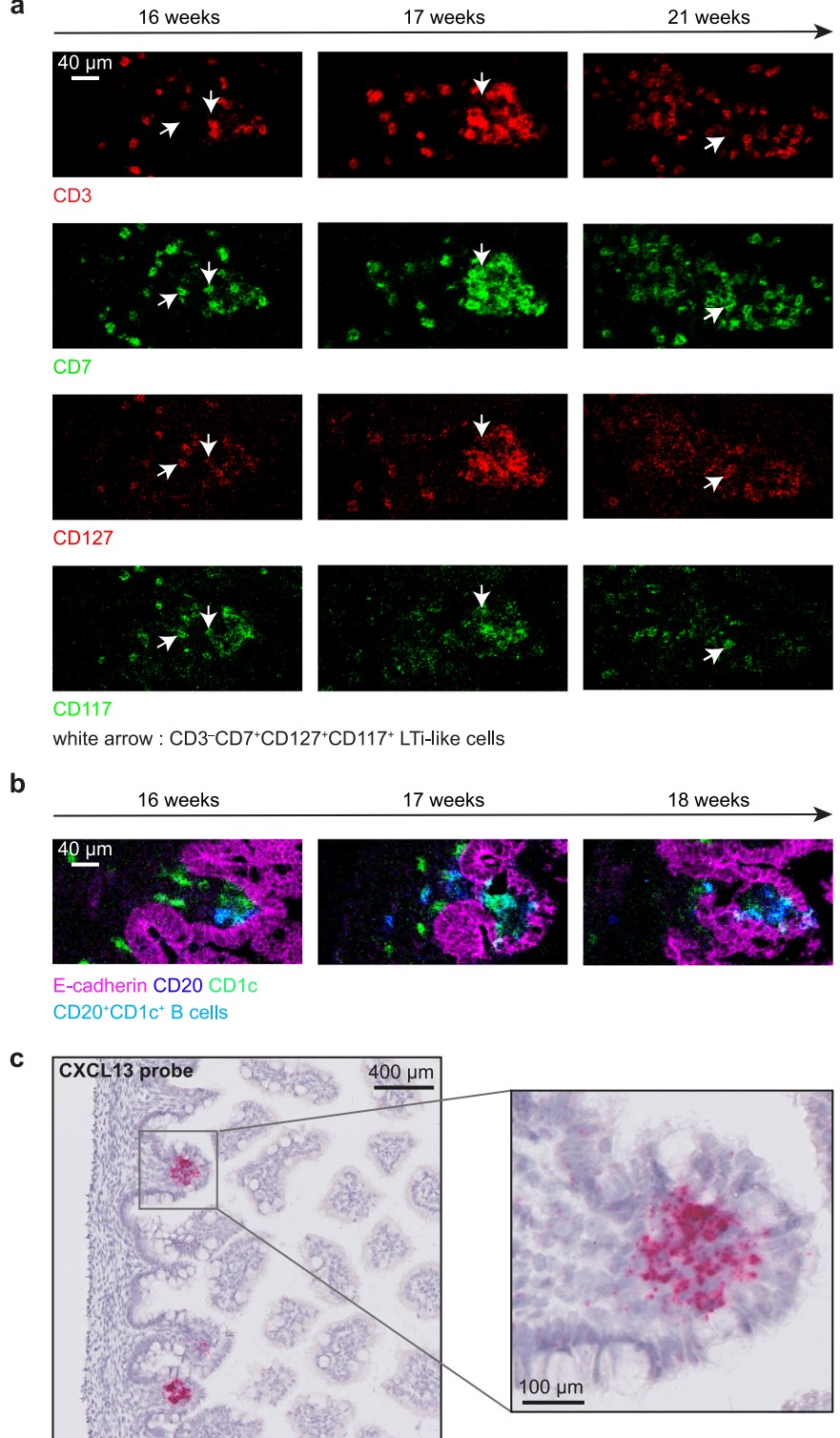

**Fig. 6 | The presence of LFs indicators in human fetal intestinal samples.**
**a** Visualization of lymphoid inducer cell-like cells (LTi-like: CD3⁻CD7⁺CD127⁺CD117⁺) within LFs. The white arrow indicates the expression of CD7, CD127 and CD117 in the absence of CD3. Scale bar, 40 μm. **b** Identification of CD20⁺CD1c⁺ B cells (magenta) within LFs. Scale bar, 40 μm. **(c)** The detection of CXCL13 mRNA in situ on a representative human fetal intestine sample from gestational week 14 by RNAscope. Data represent two independent expriments with three samples.

cells appeared to be present in the LF, while the CD163⁺ myeloid cells were abundant in the lamina propria.

Further analysis indicated the expression of CD127 and CD117 by the CD3⁻CD7⁺ ILCs, suggestive of an LTi-like phenotype (Fig. 6a). Moreover, most B cells within the LFs displayed expression of CD1c (Fig. 6b). Moreover, we analyzed the expression of CXCL13, a B cell

chemoattractant. RNAscope analysis revealed abundant CXCL13 transcripts within round structures located below the epithelium in the human fetal intestine, at a location similar to the single follicles detected by IMC (Fig. 6c). Together, this global analysis revealed the presence of similar immune subsets in LFs through gestational week 16 to 21, with a gradual increase in adaptive immune cells in time.

Together these results indicated a dynamic and distinct spatial organization of the immune compartment in LFs in the developing fetal intestine.

### A CD69⁺CD117⁺CD161⁺ CCR6⁺CD127⁺ phenotype is shared by subsets of fetal intestinal CD3⁻CD7⁺ ILCs and T cells

Next we focused our analysis on CD3⁻CD7⁺ ILCs, DN T cells, CD4⁺ T and CD8⁺ T cells in the human fetal intestine. tSNE analysis of the single-cell data revealed substantial heterogeneity in all four lymphoid cell subsets (Fig. 7a–d). As expected, a sizable proportion of CD3⁻CD7⁺ ILCs expressed CD117 (Fig. 7a). In addition, CD117-expressing cells were present in all T cell subsets (DN, CD8 and CD4) (Fig. 7b–d). Moreover, most CD117⁺ lymphoid cells were CD161⁺, CCR6⁺ and CD127⁺, while the tissue-residency marker CD69 was only present on a subset of these cells (encircled in Fig. 7b–d). Finally, small pockets of Ki-67⁺ cells were detectable in both CD69⁺ (encircled in Fig. 7b–d) and CD69⁻ (indicated by the arrow in Fig. 7b–d) counterparts of the four CD117⁺ lymphoid cell clusters, indicative of cell proliferation.

We next determined the spatial location of CD69⁺CD117⁺ CD161⁺CD127⁺ ILCs and T cells within the fetal intestine by IMC. First we performed pixel analysis to specifically detect and quantify co-expression of CD161 and CD69 inside and outside the LFs (Fig. 7e, f). Here a significantly higher proportion of CD161⁺CD69⁺ double-positive pixels was present in LFs compared to a similar surface area outside the LFs (Fig. 7e, f). Moreover, simultaneous visualization of CD3, CD7, CD4, CD8a, CD161, CD69, CD117 and CD127 in a single tissue slide revealed the presence of both CD3⁻CD7⁺ ILCs, CD4⁻CD8⁻ T cells, CD4⁺ T cells and CD8⁺ T cells displaying a CD69⁺CD117⁺CD161⁺CD127⁺ phenotype within the lymphoid follicles (Fig. 7g).

Thus, a subset of CD3⁻CD7⁺ ILC and T cells share a CD69⁺CD117⁺ CD161⁺CCR6⁺CD127⁺ phenotype, harbor cells with proliferative capacity and reside within the LFs.

### Multiple types of Ki-67⁺ proliferating cells are identified in situ

To verify the presence of Ki-67⁺ proliferating cells in each immune lineage in situ, we included a Ki-67-specific antibody in the IMC panel and applied it to frozen sections of human fetal intestinal samples (Supplementary Table 2–4). We observed Ki-67⁺ cells in all human fetal intestinal samples, in both the epithelium and the lamina propria (Fig. 8a, b). Moreover, within the LFs, Ki-67⁺ cells were also detected (Fig. 8b). Within the lamina propria, we also defined the phenotype of the Ki-67⁺CD45⁺ cells by simultaneous visualization of immune lineage markers (Fig. 8b, c). This revealed the presence of Ki-67⁺CD4⁺ T cells, Ki-67⁺CD8⁺ T cells, Ki-67⁺CD3⁻CD7⁺ ILCs, Ki-67⁺CD20⁺ B cells, Ki-67⁺CD163⁺ macrophages, and Ki-67⁺Lin⁻HLA-DR⁺ myeloid cells in the human fetal intestine (Fig. 8c, d and Supplementary Figs. 5, 6). Thus, Ki-67⁺ cells were identified in all major immune lineages in the developing fetal intestine in situ.

As most of fetal intestinal immune cells expressed CD127/IL-7Rα (Fig. 8e), we next used RNAscope analysis which revealed the presence of IL-7 transcripts in both the epithelium and lamina propria of the human fetal intestine (Fig. 8f, g, Supplementary Fig. 5b), suggesting that locally produced IL-7 could support the proliferation of the fetal intestinal immune cells.

### Fetal intestinal immune subsets display proliferation potential ex vivo

Following on the identification of proliferation-associated Ki-67⁺ cells across all major immune cell types and IL-7 mRNA expression in situ, we next aimed to determine if proliferation could be detected in fetal intestinal cells when cultured in vitro. For this purpose, proliferation assays were performed based on CellTrace™ Violet dye dilution by culturing immune cells isolated from a fetal intestinal sample in medium alone. In addition, we also assessed proliferation in the

presence of IL-7. Adult peripheral blood mononuclear cells (PBMCs) were used as a control. To analyze the cell cultures, we used a spectral flow cytometry antibody panel designed to discriminate B cells, CD3⁻CD7⁺ ILCs, DN T cells, CD8⁺ T cells, and CD4⁺ T cells (including CD25⁺ Treg-like, CD117⁻CD161⁺ and CD117⁺CD161⁺ subsets).

All subsets were identified in both culture conditions (Fig. 9a), indicating the presence of stable immune phenotypes in time. In the absence of IL-7, the dilution of CellTrace™ Violet dye revealed cellular division accompanied by the expression of Ki-67 in all fetal intestinal immune subsets (Fig. 9b). This was enhanced by the addition of IL-7 (Fig. 9b). In contrast, proliferation was hardly observed in the cultures of the PBMC sample (Fig. 9c, Supplementary Fig. 7). A similar analysis of fetal immune cells from samples of gestational weeks 16, 18, 20 and 21 likewise revealed spontaneous proliferation of fetal intestinal immune subsets (Fig. 9d). In almost all cell cultures we observed that addition of IL-7 resulted in an increase of CTV-diluted cells, pointing towards enhanced proliferation.

To determine functional characteristics of the cultured fetal intestinal immune cells, we analyzed the expression of CD40L, granzyme B, TNF-α, IFNγ, IL-2 and IL-17A by flow cytometry (Supplementary Fig. 8). In both the presence and absence of IL-7, a substantial proportion of CD3⁻CD7⁺ ILCs, DN T cells and CD4⁺ T cells expressed CD40L after 48 hours of culture, while this was much lower on CD8⁺ T cells (Supplementary Fig. 8a–d). Granzyme B was detected in CD3⁻CD7⁺ ILCs, DN T cells and CD8⁺ T cells, while intracellular levels of TNF-α, IFNγ, IL-2 and IL-17A were low in all cell populations in both culture conditions (Supplementary Fig. 8a–d). Furthermore, in the presence of IL-7 the median fluorescence intensities (MFI) of CD40L expression increased substantially in CD3⁻CD7⁺ ILCs, DN T cells and CD4⁺ T cells (Fig. 9e, Supplementary Fig. 8e). Moreover, higher frequencies of CD40L⁺granzyme B⁺ ILCs and T cells were detected after 48 hours of culture in the presence of IL-7 (Fig. 9f, Supplementary Fig. 8e).

Thus, all immune subsets in the fetal intestine contain cells that give rise to progeny in vitro, in particular in the presence of IL-7, which is accompanied by upregulation of CD40L and granzyme B expression.

## Discussion

The human intestine harbors a significant fraction of the body's immune cells[20–22] that controls local responses to the microbiota and food antigens to maintain homeostasis[23]. We and others have described the presence of both innate and adaptive cells with complex immune phenotypes in the human fetal intestine[14,15,24,25]. Moreover, direct comparison of fetal and infantile samples has revealed substantial differences in immune composition in both the adaptive and innate compartment[16]. Yet little is known about the temporospatial development of the fetal intestinal immune system.

Here, we have used spectral cytometry and imaging mass cytometry (IMC) to explore the composition and distribution of immune cells in the human fetal intestine across the second trimester (14 to 22 weeks of gestation). At gestational week 14–15, the immune compartment was mostly composed of myeloid cells and CD3⁻CD7⁺ ILCs (Figs. 1 and 2), which corresponds with observations in other fetal tissues, where these innate cell subsets appear earlier than their adaptive counterparts[15]. In suspension, most myeloid cells co-expressed CD11c, HLA-DR and CD45RO, with some cells expressing CD1c or CD163. IMC confirmed the abundant presence of Lin⁻HLA-DR⁺ and CD163⁺HLA-DR⁻ cells in different anatomical locations. While Lin⁻HLA-DR⁺ cells were close to the epithelium in the lamina propria, CD163⁺HLA-DR⁻ cells were found both in the LP as well as in the submucosa (Fig. 5). Previously we have provided evidence for the presence of several CD3⁻CD7⁺ ILC subsets in the fetal intestine, including ILC3[24]. In agreement, we now observed three phenotypically distinct clusters of CD3⁻CD7⁺ ILCs, one of which displayed features of ILC3/lymphoid tissue inducer cells (LTi; CD3⁻CD7⁺CD117⁺CD161⁺CCR6⁺

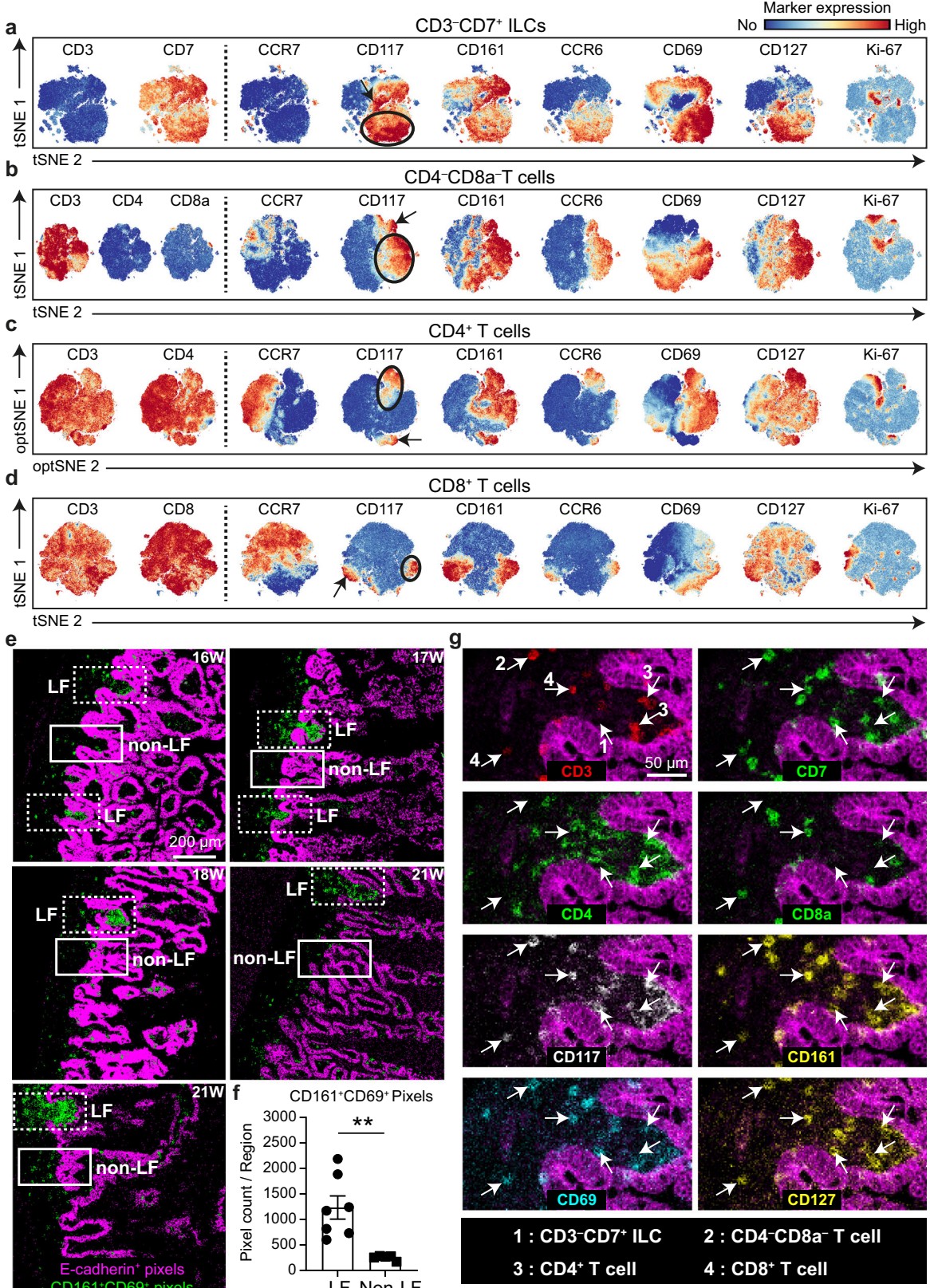

CD127+RORγt+). These results are in line with a recent study in which Elmentaite et al. used single-cell RNA sequencing to map the cell types in the human intestine[26]. They reported that LTi-like NCR+ and NCR-ILC3s were particularly abundant in the fetal intestine compared to both pediatric and adult samples. Moreover, these were located in proximity to CXCL13-expressing lymphoid tissue organizer-like stromal cells, in line with their role in lymphoid tissue formation in the fetal gut[26]. By means of RNAscope assays, we also identified the presence of CXCL13 transcripts in the human fetal intestine, matching the shape, location and distribution of lymphoid follicles identified by IMC (Fig. 6c). CXCL13 can be produced by mesenchymal lymphoid tissue organizer cells and attracts LTi cells towards the parenchyma of the

**Fig. 7 | Subsets of T cells and ILCs display a highly similar CD69$^+$CD161$^+$CD117$^+$CCR6$^+$CD127$^+$ phenotype. a–d** tSNE analysis of the CD3$^-$CD7$^+$ ILCs cells ($2.4 \times 10^5$ cells), CD4$^-$CD8$^-$ T cells ($1.9 \times 10^5$ cells), CD4$^+$ T cells ($1.2 \times 10^6$) and CD8$^+$ T cells ($3.1 \times 10^5$ cells) from 12 human fetal intestinal samples from gestational week 14 through 22. Colors represent the relative expression of the indicated immune markers. CD69$^+$CD161$^+$CD117$^+$CCR6$^+$CD127$^+$ cells are encircled, while their CD69$^-$ counterparts are indicated by arrows. Data represent three independent experiments. **e** CD161$^+$CD69$^+$ immune cells (green) and epithelium (pink) were visualized by pixel analysis in the human fetal intestine from different gestational ages. LFs are boxed by dashed lines, while similar area's devoid of LFs

(non-LFs) are boxed by solid line. Scale bar, 200 µm. **f** CD161$^+$CD69$^+$ pixels were quantified within LFs ($n = 7$) and non-LF ($n = 5$). Error bars indicate mean ± s.e.m. ** $P < 0.01$, Mann-Whitney test with one-tailed for comparisons. Data were from three independent experiments. **g** The combination of the immune cell markers indicated was used to visualize CD69$^+$CD161$^+$CD117$^+$CD127$^+$ CD3$^-$CD7$^+$ ILC (indicated by arrow 1), CD69$^+$CD161$^+$CD117$^+$CD127$^+$ CD4$^-$CD8$^-$ T cell (arrow 2), CD69$^+$CD161$^+$CD117$^+$CD127$^+$ CD4$^+$ T cell (arrow 3) and CD69$^+$CD161$^+$CD117$^+$CD127$^+$ CD8$^+$ T cell (arrow 4) in the LF of a human fetal intestinal sample in situ. The images were representative of three independent experiments. Scale bar, 50 µm.

LNs anlagen in the fetal stage[27]. CXCL13 is also known to be involved in the recruitment of B cells[28]. Together, this suggests a role for CXCL13 in the formation of the lymphoid follicles in the fetal intestine. Remarkably, ILC3s in tissues from patients with Crohn's disease matched fetal NCR$^+$ ILC3s with more than 60% probability, which suggests fetal lymphoid tissue development programs could be implicated in the Crohn's disease pathogenesis[26].

From 17 weeks onwards, B and T cells were the most abundant cell subsets and the composition of the fetal intestinal immune compartment remained relatively stable (Figs. 1 and 2). B cells were divided into two clusters. The most prominent of these clusters were composed of CD20$^+$HLA-DR$^+$ B cells that co-expressed CCR6 and CD1c. In humans, CD1c is expressed by mantle zone B-cells of the tonsil, lymph node and spleen, and marginal zone B (MZB) cells of the spleen[29]. Recently, high expression of CD1c was described in a population of transitional B cells that migrate to the gut and later give rise to MZB cells[30]. MZB cells are strategically located at the interface between the circulation and the white pulp of the spleen, providing the first line of defense by rapidly producing IgM and IgG antibodies in response to infections[31]. They can also be found in the subepithelial dome of intestinal Peyer's patches and produce IgM, IgG and IgA antibodies to commensal antigens[31]. IMC analysis of human fetal intestines revealed the accumulation of fetal CD1c$^+$ B cells in lymphoid follicles, in close contact with CD3$^-$CD7$^+$ ILCs, T cells and HLA-DR$^+$ cells (Figs. 5 and 6). As transitional B cells must migrate to the gut to give rise to mature clonally-diverse MZB[32], it is thus plausible that the fetal LFs are the maturation site of fetal MZB precursors where they are in close contact with T cells and HLA-DR$^+$ myeloid cells.

We previously described that substantial heterogeneity within the CD4$^+$ T cell compartment was already present at 17 weeks[14]. Moreover, substantial evidence has been presented indicating that the majority of fetal intestinal CD4$^+$ T cells display a memory phenotype[14–16]. Likewise, fetal intestinal CD8$^+$ T cells can display a memory phenotype. It has been speculated that this points to *in utero* exposure to foreign antigens. In agreement, Mishara et al. recently provided evidence for the presence of bacterial species in fetal samples[33]. However, many have argued that the detection of microbial species in such samples is caused by contamination and this debate may not easily be settled. In any case, microbial presence in the fetus is sparse at best[33] and may not be sufficient to explain the abundance of memory T cells in the human fetal intestine samples. Alternatively, memory formation could be triggered by exposure to proteinous and non-proteinous antigens that gain access to the amniotic fluid, possibly derived from the maternal microbiota. Our previous results indicated that memory formation is associated with several signaling pathways, including that of the T cell receptor, supporting the notion that exposure to foreign antigens is at least partly responsible for the observed memory formation[14]. However, our current results indicate that the memory phenotype may also be associated with the expansion of adaptive immune cells, either in the developing fetal intestine or an intermediate tissue, perhaps analogous to the thymus where the majority of double-negative and single-positive thymocytes express CD45RO and display proliferative potential. Further research will be required to explore this option.

Of note, fetal mesenteric lymph nodes at 20 gestational weeks contains T cells with both naïve and memory-like phenotype[34].

As observed before[14], a significant proportion of CD4$^+$ T cells expressed CD161 and lacked expression of CCR7, and a similar pattern was observed for CD3$^-$CD7$^+$ ILCs, DN and CD8α$^+$ T cells (Fig. 7), potentially reflecting the shared transcriptional program observed in adult human CD161$^+$ circulatory CD4$^+$, CD8α$^+$ (including MAIT cells) and TCRγδ$^+$ T cells[35]. Further analysis of the CD161$^+$ cell subsets revealed that two clusters of CD117$^+$CCR6$^+$CD127$^+$ cells were present that either lacked or expressed CD69 within CD3$^-$CD7$^+$ ILCs, DN, CD4$^+$ and CD8α$^+$ T cells (Fig. 7a–d). Moreover, CD3$^-$CD7$^+$ ILCs, DN, CD4$^+$ and CD8α$^+$ T cells displaying this CD69$^+$CD117$^+$CCR6$^+$CD127$^+$ phenotype were identified in LFs by IMC (Fig. 7e), and pixel analysis confirmed the enrichment of CD161$^+$CD69$^+$ pixels in areas where LFs were present (Fig. 7f). Together, the shared phenotype of the CD69$^+$CD117$^+$CD161$^+$ CCR6$^+$CD127$^+$ cells across subsets and their localization in the LF may indicate a common function in LF development, a matter that will be investigated in future studies. So far, CD161 has been associated with IFNγ and IL-17A in adult CD4$^+$ T cells[36], or IFNγ production in adult CD4$^+$, CD8α$^+$ and TCRγδ$^+$ T cells[35]. Interestingly, Cosmi et al. revealed the presence of naïve CD161$^+$CD4$^+$ T precursor cells in cord blood and postnatal thymus, that expressed RORγt and CCR6 and could differentiate into IL-17A/IFNγ-producing cells[36]. Such committed thymus-derived naïve CD161$^+$CD4$^+$ T cell precursors may enter the circulation during fetal development and home to the intestine.

While the presence of Ki-67$^+$ in the human fetal intestine has been noted before, detailed analysis of fetal samples allowed us to identify stable pockets of Ki-67$^+$ cells in all immune cell subsets throughout a major part of the second trimester. This result is compatible with the sustained proliferation of immune-subset committed cells in time (Figs. 1 and 2 and Supplementary Fig. 1–3). Ki-67 is expressed at variable levels throughout the cell cycle, reaching its maximum during the S-M phases and decreasing via degradation in G1/G0[37]. High levels of Ki-67 expression, which would be detectable by IMC, suggest lack of quiescence and rapid progression through the cell cycle. This is supported by a re-analysis of our previously generated single-cell RNA sequencing data[14] where a subset of the fetal intestinal CD4$^+$ T cells were found to co-express *CCNB2*, *CDK1* and *MKI67* (Fig. 4b), genes associated with cell cycle. Moreover, spontaneous and IL-7-induced proliferation of fetal intestinal cells were observed in vitro and IMC confirmed the presence of Ki-67$^+$ cells with different phenotypes both in LFs as well as scattered along the LP (Fig. 8 and Supplementary Figs. 5, 6). Of note, in the infant intestinal tract Ki-67$^+$ staining was only detectable within isolated LFs and the epithelium, but not in the lamina propria[38]. Moreover, TREC analysis showed that T cells in the fetal intestine underwent an average of 3.5 (week 14) to 6.5 (week 22) divisions (Fig. 4a). Analysis of the expression of genes involved in TCR rearrangement (*RAG1*, *RAG2*, *DNTT* and *LIG4*) revealed that only a few cells expressed *RAG1* and *LIG4*, while *RAG2* and *DNTT* were not expressed (Fig. 4d), arguing against extrathymic T cell development in the fetal intestine.

During the second trimester, the intestine not only doubles in length[39], but also villus and microvillus formation drive an even greater increase in the intestinal surface area[40]. Thus, immune cell colonization

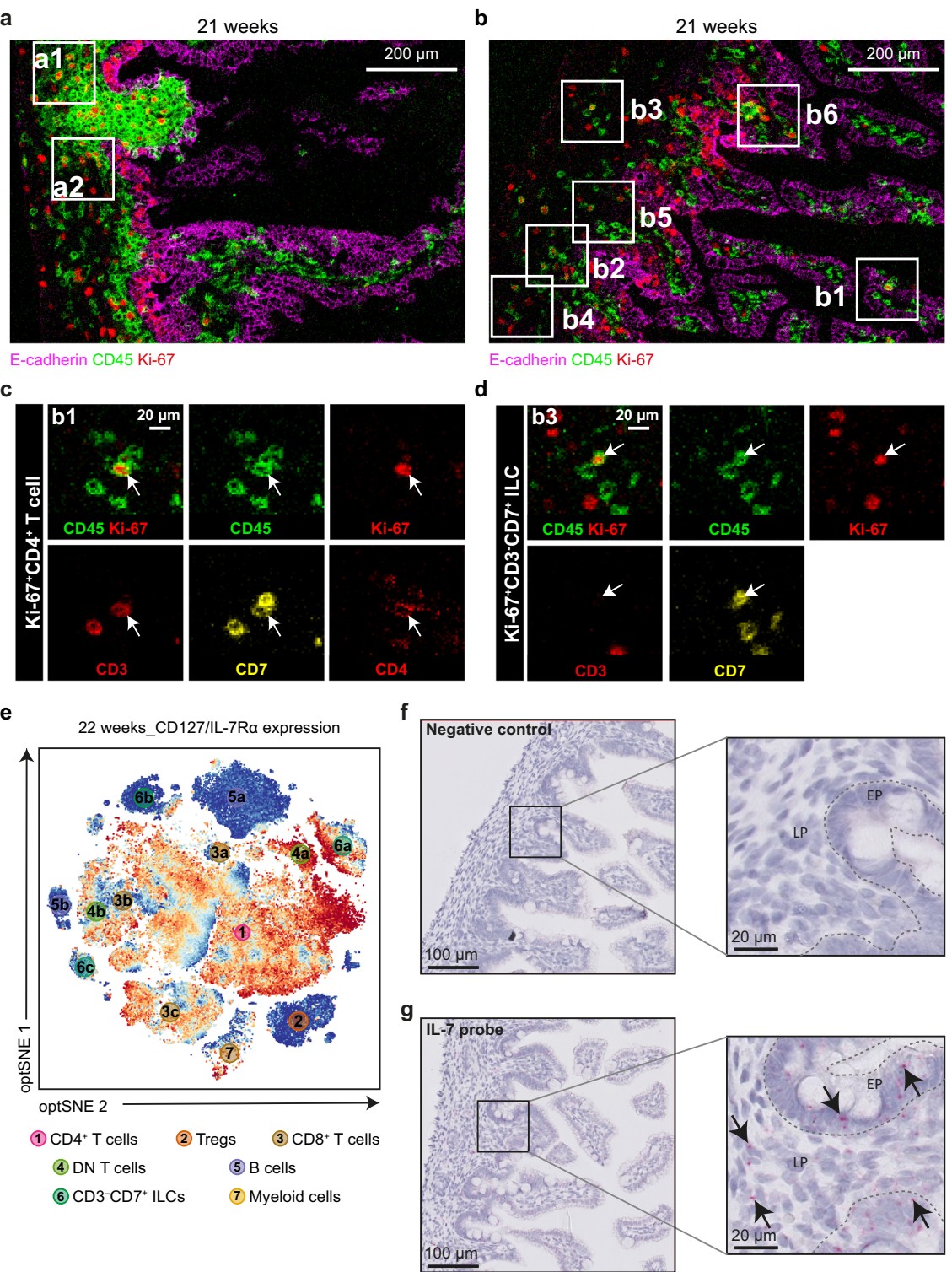

**Fig. 8 | Visualization of Ki-67 and IL-7 expression in situ. a** Visualization of Ki-67+CD45+ cells in a LF by overlay of Ki-67 and CD45 on a representative human fetal intestinal sample from gestational week 21. Scale bar, 200 μm. **b** Visualization of Ki-67+CD45+ cells in the lamina propria by overlay of Ki-67 and CD45 on an additional human fetal intestinal sample from gestational week 21. Scale bar, 200 μm. **c, d** The combination of Ki-67 and immune cell markers was used to visualize **(b1)** Ki-67+CD4+ T cells, **(b2)** Ki-67+CD3−CD7+ ILCs in human fetal intestinal samples as indicated by white arrows. Scale bar, 20 μm. **e** The expression of CD127 (IL-Rα) on major immune subsets from the human fetal intestinal sample from gestational week 22. **f, g** The detection of IL-7 mRNA by RNAscope assays on the human fetal intestine: (**f**) *DapB* as negative control (**g**) IL-7 probe. Data represent two independent expriments with three samples. EP epithelium, LP lamina propria.

of the LFs and lamina propria requires an ever-increasing number of immune cells. This can occur via continuous input from the fetal liver, thymus and bone marrow, and/or local proliferation. Given the constant proportion of Ki-67+ cells in the fetal intestine and their presence in every immune subset throughout most of the second trimester, the most likely scenario is that fetal T cells develop in the thymus and then migrate to the periphery where they locally expand. Based on both Ki-67 expression and division history, we can envision two scenarios.

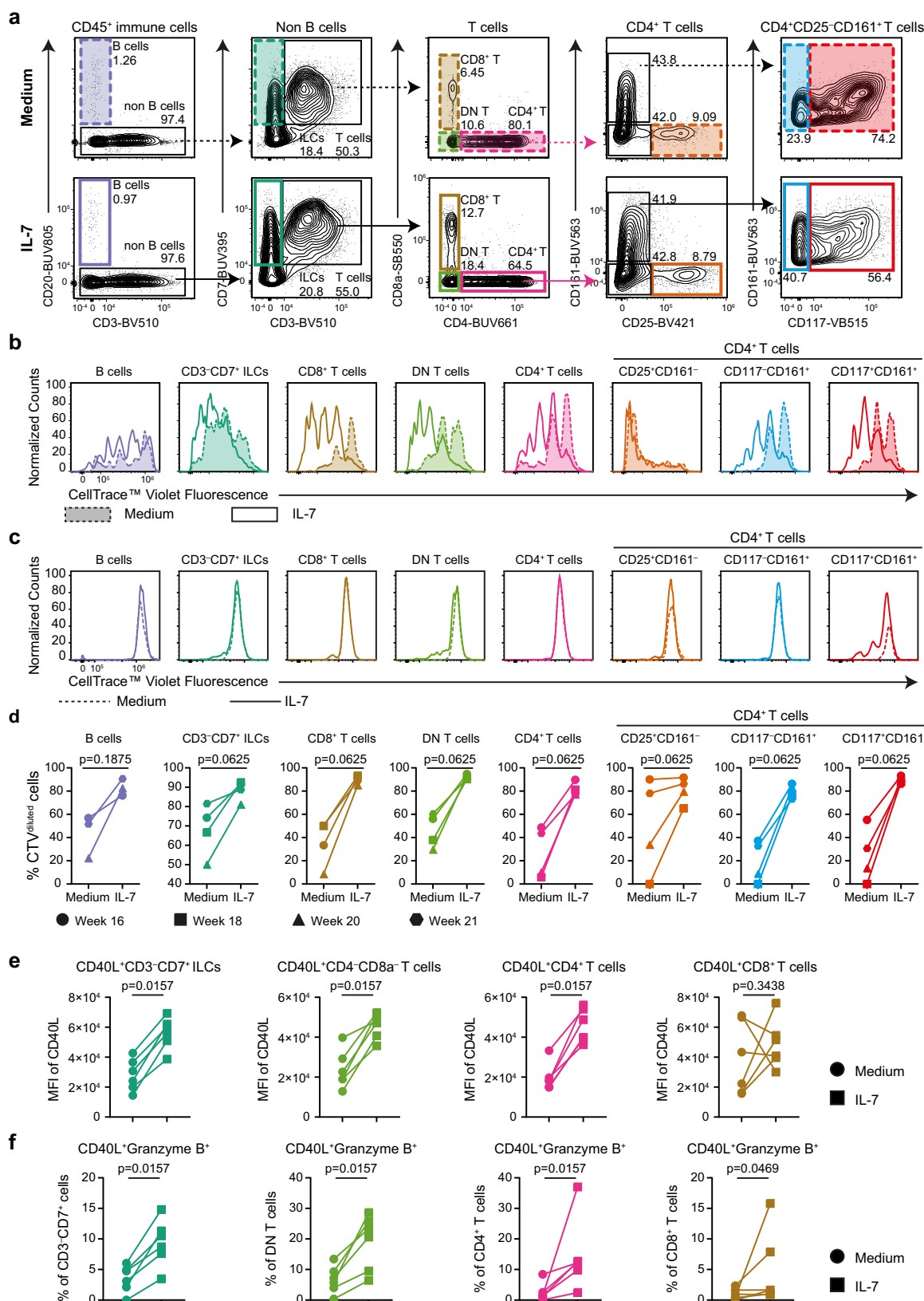

In the first scenario, differentiated immune cells - including T cells-travel from the primary lymphoid organs to secondary lymphoid organs or intermediate tissues, where they undergo a few rounds of proliferation and then travel to the intestine. Given that Ki-67 is detectable at stable levels, the intestine would receive recently divided cells at a regular rate throughout the second trimester. In the second scenario, differentiated immune cells would reach the intestine at a regular rate during the second trimester, where they would receive local homeostatic proliferation signals and undergo a few rounds of division. In favor of the latter, we have detected IL-7 transcripts in the human fetal intestine, both in the epithelium and the lamina propria (Fig. 8g).

**Fig. 9 | Fetal intestinal samples harbor cells with proliferative capacity in vitro.** **a** Biaxial plots showing the gating strategy to identify immune subsets in a human fetal intestinal sample from gestational week 16 cultured in medium alone or medium with IL-7 for 4 days. The colored gates indicate the identified immune subsets. Data represent four independent experiments. **b** Histogram showing CellTrace™ Violet dye dilution in the indicated immune subsets from the human fetal intestine. Dashed lines represent the fetal intestinal sample cultured in medium, while solid lines represents the fetal intestinal sample cultured in medium with IL-7. **c** Histogram showing CellTrace™ Violet dye dilution by the indicated immune subsets from a PBMCs sample. Dashed lines represent the PBMCs cultured in medium, while solid lines represents the PBMCs cultured in medium with IL-7.

**d** CTV$^{diluted}$ cells of the immune subset indicated from human fetal intestinal samples in both culture conditions. Data were from four independent experiments ($n = 4$). Error bars indicate mean ± s.e.m. Wilcoxon signed-rank test with one-tailed for comparisons. **e** Comparison of the mean MFI values of CD40L expression between culture in medium alone and in the presence of IL-7. Data were from four independent experiments ($n = 6$). **f** Comparison of the percentages of CD40L$^+$granzyme B$^+$ expressing cells within CD3$^-$CD7$^+$ ILCs, CD4$^-$CD8$^-$ T cells, CD4$^+$ T cells and CD8$^+$ T cells in both culture conditions. Data were from four independent experiments ($n = 6$). Error bars indicate mean ± s.e.m. Wilcoxon signed-rank test with one-tailed for comparisons.

The situation of proliferating immune cells in a growing intestine could be paralleled to the filling of niches in a lymphopenic host through homeostatic proliferation, likely in the absence of antigen. It could be envisioned that as the organ grows, new stromal niches are generated that provide signals for proliferation. In support of this idea, it has been shown that the neonatal environment in mice is functionally lymphopenic, where transferred CD4$^+$ T cells undergo proliferation[41]. In immune-deficient hosts, lymphopenia-induced proliferation is driven by the increased availability of IL-7[41–44]. We detected CD127 (IL-7Rα) on most fetal human lymphoid immune cells, except Tregs, B cells and some CD3$^-$CD7$^+$ ILC subsets (Fig. 1 and Supplementary Figs. 1–3). Unlike peripheral immune cells from healthy adults, fetal cells responded to IL-7 with vigorous proliferation after four days in vitro. This was accompanied by upregulation of CD40L and granzyme B in CD3$^-$CD7$^+$ ILCs, CD4$^+$ T cells, CD8αβ$^+$ T cells and DN T cells, in the absence of concurrent cytokine production (Fig. 9e, f, Supplementary Fig. 8). In healthy adults, IL-7-induced proliferation was found to be restricted to CD31$^+$ recent thymic emigrants (RTE)[45]. Fetal intestinal CD4$^+$ T cells may be RTEs, explaining their increased sensitivity towards IL-7. In mice, IL-7 is produced by stromal cells[46]. Further work will be required to elucidate the source of IL-7 in the human fetal intestine.

In summary, we identified a stable presence of Ki-67$^+$ cells in all major immune subsets of human fetal intestines during the second trimester, and IL-7-enhanced proliferation in vitro. Through IMC, we visualized the presence and development of fetal LFs throughout the second trimester and characterized the shared phenotype of CD69$^+$CD117$^+$CD161$^+$CCR6$^+$CD127$^+$ by fetal intestinal T cells and CD3$^-$CD7$^+$ ILCs frequently located within such LFs. Overall, these observations indicate the presence of immune subset-committed cells capable of local proliferation, contributing to the population size and development of an organized immune compartment throughout the 2$^{nd}$ trimester in the human fetal intestine.

## Methods

### Sample processing and cell isolation

The human fetal material used in this work was obtained from elective abortions (without medical indication) with signed informed consent from all donors. The pediatric and adult intestinal samples were from uninflamed regions of patients. The work described here was reviewed and approved by the Medical Ethical Committee of Leiden University Medical Centre (P08.087). The gestational age ranged from 14 to 22 weeks. The small intestine and colon were separated from mesentery, cut into small pieces, embedded in optimal cutting temperature compound, snap-frozen in isopentane and stored at −80 °C. In addition, intestinal pieces were fixed in formalin and embedded in paraffin (FFPE tissue). The remainder of the material was used for single-cell isolation as described previously[24]. Briefly, after clearing of meconium, fetal intestines were cut into small fragments in a petri dish, then incubated in 15 mL 1 mM dithiothreitol (Fluka) for 10 min twice at room temperature, and then incubated with 1 mM ethylenediaminetetraacetic acid (Merck) for 1 hour twice at 37 °C under rotation to acquire intraepithelial lymphocytes (IELs). To obtain single-cell suspensions

from the lamina propria, the intestines fragments were enzymatically digested with 10 U/mL collagenase IV (Worthington) and 200 µl/mL DNAseI grade II (Roche Diagnostics) in 15 mL of Hank's balanced salt solution (ThermoFisher Scientific) overnight at 37 °C. After incubation, the cell preparations were filtered through a 70 µm cell strainer (Corning) followed by washing of the cells with Iscove's Modified Dulbecco's Medium (IMDM, Lonza). Isolated cells were then purified with a Percoll gradient (GE Healthcare). Purified cells were cryopreserved in liquid nitrogen until time of analysis in 90% FCS and 10% dimethyl sulfoxide (DMSO) (Merck). All experiments were conducted in accordance with local ethical guidelines and the principles of the Declaration of Helsinki.

### Spectral flow cytometry immunophenotypic studies

The 26-antibody flow cytometry-based panel was developed for in-depth immunophenotyping of the major cell subsets present in the human fetal intestine through time. In total, 3 experiments were performed for immunophenotypic studies of 28 human fetal intestinal samples. Antibodies used for spectral flow cytometry with a 5-laser Cytek® Aurora are listed in Supplementary Table 1. For surface staining, single-cell suspensions of fetal intestinal samples were incubated with fluorochrome-conjugated antibodies and human Fc block (BioLegend) for 30 min at 4 °C. After washing, cells of samples were then fixed/permeabilized using Foxp3 Staining Buffer Set, according to manufacturer's instructions (ThermoFisher). For intracellular staining, the fixed/permeabilized cells were incubated with the antibodies for 45 min at 4 °C, followed by washing of the cells with permeabilization buffer. Then the stained cells were resuspended. Reference samples were incorporated and individually stained by UltraComp eBeads™ Compensation Beads (ThermoFisher), PBMCs or cells from human tonsils. After the completion of the sample preparation, the samples were immediately acquired using a 5-laser Cytek® Aurora (Cytek® Biosciences). Data were analyzed to check quality with FlowJo software version 10.6 (Tree Star Inc). We utilized OMIQ to perform the high-dimensional analysis for human fetal intestinal samples (https://www.omiq.ai/).

### Imaging mass cytometric immunostaining, acquisition and analysis

Antibodies and staining procedures used on the human fetal intestine for IMC were optimized from the antibody panel and protocol developed in a previous publication[18]. IMC immunostaining was performed on frozen human fetal intestine samples in three independent experiments, and antibodies used for each IMC are listed in Supplementary Table 2-4. The carrier-free formulations of antibodies were conjugated to lanthanide metals using the MaxPar Antibody Labeling Kit (Fluidigm) following the manufacturer's instructions. We dried the tissue section for 1 h at 60 °C, followed by fixation with paraformaldehyde (PFA) for 5 min at RT, then cold methanol for 5 min at −20 °C. Then, the samples were incubated with the antibody panel at 4 °C overnight. Next, tissue acquisition was performed on a Helios time-of-flight mass cytometer coupled to a Hyperion Imaging System (Fluidigm). All IMC operations were performed as described using the

Hyperion Imaging System (Fluidigm). All raw data were analyzed for marker intensity based on the maximum signal threshold, defined at the 98th percentile of all pixels in a single ROI using Fluidigm MCD™ viewer (v1.0.560.2).

We developed the computational tool "Cytosplore Imaging" to analyze multiple markers simultaneously to perform pixel analysis for IMC data. Cytosplore Imaging facilitates the complete exploration pipeline in an integrated manner in a ROI (Supplementary Fig. 4). Data analysis in Cytosplore Imaging included the following steps: We applied the arcsin transformation with a cofactor of five upon loading the data sets (ome.TIFF files) exported from Fluidigm MCD™ viewer. Next, we applied an HSNE analysis[19] on $2.5 \times 10^5$ pixels from an ROI and defined the markers used for the similarity computation: CD45, D2-40, Collagen I, αSMA, CD31, E-cadherin, CD123, CD7, CD163, CD20, CD11c, CD161, Ki-67, HLA-DR, CD45RA, CD3, CD57, vimentin and CD56. Based on marker expression (Supplementary Fig. 4a), we could distinguish tissue pixels from background pixels and projected each population on the imaging viewer (Supplementary Fig. 4b). Next, we selected tissue pixels and zoomed in on these through a new HSNE analysis to visualize the structural components and identify immune cells located in the human fetal intestine (Supplementary Fig. 4c, d). Finally, we focused on CD45$^{+/dim}$ pixels and performed a t-SNE analysis clustering for the immune markers: CD45, CD3, CD7, CD20, HLA-DR, CD163 (Supplementary Fig. 4e). Using this approach, we could visualize T cells, CD3$^-$CD7$^+$ cells, B cells, CD163$^+$ macrophages, Lin$^-$HLA-DR$^+$ myeloid cells and E-cadherin$^+$ epithelial cells simultaneously in a single region (Fig. 5f).

### RNAscope assay procedure for RNA detection

RNAscope (ACD, BioTechne) was applied to fetal intestinal FFPE tissue according to manufacturer's instructions[47]. Firstly, 5μm FFPE tissue sections were deparaffinized in xylene, followed by dehydration in an ethanol series. After deparaffinized the slides were dried and RNAscope® Hydrogen Peroxide was applied to cover the entire slides, incubated for 10 min at RT, and washed with distilled water. Then, the slides were put in a container containing RNAscope® 1× Target Retrieval Reagent for 30 min at 99 °C. After removing the slides from the container, the slides were rinsed for 15 second in distilled water, washed 3 min in 100% alcohol and dried in an incubator at 60 °C. The processed tissue sections in the slides were circled with an Immedge™ hydrophobic barrier pen, and tissue sections were covered by RNAscope® Protease Plus for 30 min at 40 °C, followed by washing 3–5 times in distilled water. Subsequently, the IL-7 and CXCL13 probes were added followed by incubation for 2 hour at 40 °C after which the slides were washed in 1× Wash Buffer for 2 min twice at RT. Next, signal amplification was performed following the protocol from RNAscope® 2.5 HD Detection Reagents. After the hybridization steps, slides were washed with wash buffer 3 times at RT. Lastly, tissue sections were incubated with RED solution for 10 min at RT, followed by a counterstaining with hematoxylin. Assays using FFPE samples were typically performed in parallel with *PPIB* probe as positive control and *DapB* probe as negative control. The probed regions from target genes are listed in Supplementary Table 5.

### Cell proliferation assays by spectral flow cytometry

After thawing, single-cell suspensions of fetal intestinal samples were kept in a pre-warmed PBS buffer. We prepared CellTrace™ Violet stock solution in 5 mM by DMSO (ThermoFisher). Cells were incubated with 2 mL of 1250 nM CellTrace™ Violet dye in PBS buffer for 8 min at 37 °C. Then 10 mL IMDM (Lonza) supplemented with 10% FCS was added and incubated at RT for 5 min. Next, cells were spun down at 1500 rpm for 10 min, and cells were resuspended in 6 mL prewarmed IMDM (Lonza) supplemented with 10% FCS medium at RT for 10 min before seeding in a 96-well cell culture plate. Cells were maintained in culture medium (IMDM supplemented with 10% human serum) or in culture medium containing 25 ng/mL IL-7 (Peprotech) for 4 days. The phenotype of generated progeny as well as proliferation were determined by flow cytometry. Details on antibodies used are listed in Supplementary Table 6. After the cell surface and intracellular staining as described before, cells were acquired on a 5-laser Cytek® Aurora (Cytek® Biosciences). Four independent experiments were performed for four fetal intestinal samples, and PBMC samples were included as a control. Data was analyzed with FlowJo software version 10.6 (Tree Star Inc).

### TREC analysis

DNA was extracted from fetal cells and used in a the δREC-ψJα TREC analysis. 50 ng total DNA was used in qPCR experiments as described by van der Weerd et al.[48], using the following primers and probes on a TaqMan lightcycler: sj TREC ψJα-δREC_F:CCATGCTGACACCTCTGGTT; sjψJα-δREC_R: TCGTGAGAACGGTGAATGAAG; sj TRECψJαδREC_T: [FAM]CACGGTGATGCATAGGCACCTGC[TAM]probe; cjψJαδREC_F:AA GCAACATCACTCTGTGTCTAGCAC;cjψJα-δREC_R: AATTCTGCCAAAT ATTACTTACTTGCTGAG; cjψJα-δREC_T [FAM]CCAGAGGTGCGGGCC CCA[TAM] probe and genomic albumin primers/probe ALB_F: TGA AACATACGTTCCCAAAGAGTTT ALB_R: CTCTCCTTCTCAGAAAGTGT GCATAT; ALB_T [FAM]TGCTGAAACATTCACCTTCCATGCAGA[TAM]. Data were calculated as Δ Ct and number of divisions reported as described[49].

### Cytokine production analysis by spectral Flow cytometry

After staining with CellTrace™ Violet dye as described before, single-cell suspensions of fetal intestinal samples and PBMCs samples were seeded into 96-well round-bottom plates in IMDM/L-glutamine medium (Lonza) complemented with 10% human serum with or without 25 ng/mL IL-7 for 2 days at 37 °C. Brefeldin A solution (10 ng/mL, Sigma-Aldrich) was added for the final 5 hours. PBMCs stimulated with anti-CD3/anti-CD28 agonist antibodies (2.5 μg/ml each, BioLegend) for the final 6 hours were included as a control. The flow cytometry antibody panels were designed to detect granzyme B, IL-2, IL-17A, CD40L, IFNγ and TNF-α production in identified immune subsets at day 2 by the Cytek® Aurora (BD Biosciences). Four independent experiments were performed. Details on the flow cytometry antibodies are available in Supplementary Table 7. Data was analyzed with FlowJo software version 10.6 (Tree Star Inc). All statistics were analyzed using GraphPad Prism8 software.

### Statistics

Results are shown as mean ± s.e.m. The statistics tests used were Wilcoxon matched-pairs signed-ranks test, and Mann-Whitney test for comparison. $P < 0.05$ was considered to be statistically significant. All statistics were analyzed using GraphPad Prism8 software.

### Reporting summary

Further information on research design is available in the Nature Portfolio Reporting Summary linked to this article.

## Data availability

Source data are provided as a Source Data file. Single-cell RNA-seq data is available via Gene Expression Omnibus accession code GSE122846. The flow cytometry data generated in this study have been deposited in Flow Repository (http://flowrepository.org/id/FR-FCM-Z5MB). Imaging mass cytometry data are deposited at Mendeley Data (https://data.mendeley.com/datasets/m4vr79wsjs). Source data are provided with this paper.

## Code availability

The installer of Cytosplore Imaging and the PDF file of UserGuide are provided in https://sec.lumc.nl/mtg-viewer/imaging/win/se_3.3.2/Cytosplore_Imaging_SE_3.3.2.zip.

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

## Acknowledgements

We would like to thank the staff of the Center for Contraception, Abortion and Sexuality (Leiden and The Hague) for collection and provision of fetal material used in this work as well as the donors. We thank all operators in the Flow Cytometry Core Facility of the LUMC for technical assistance with measurements on Helios and Hyperion. We thank Marieke E. Ijsselsteijn for providing help and suggestions with IMC experiments, Laura F. Ouboter, Andrea E. van der Meulen-de Jong and Caroline R. Meijer for providing pediatric and adult intestinal samples. This research was supported by the China Scholarship Council (NG, NL and LJ). SMCSL was supported by Novo Nordisk Foundation (Renew NNF21CC0073729). FK was supported by BIOMAP (Bio-markers in Atopic Dermatitis and Psoriasis), a project funded by the Innovative Medicines Initiative 2 Joint Undertaking under grant agreement no. 821511 and by the collaboration project TIMID (LSHM18057-SGF) financed by the PPP allowance made available by Top Sector Life Sciences & Health to Samenwerkende Gezondheidsfondsen (SGF) to stimulate public-private partnerships and co-financing by health foundations that are part of the SGF.

## Author contributions

N.G., M.F.P. and F.K. conceived the study and wrote the manuscript. N.G. performed most experiments with the help of J.L., N.L., Q.J, M.S., and A.A.V. Moreover, N.G., J.L., N.L., V.v.U., F.S., N.F.C.C.d.M. analyzed the data. S.M.C.d.S.L. collected and isolated the fetal intestine material. J.E. and B.L. set up the package of Cytosplore imaging for pixel analysis. All authors discussed the results and commented on the manuscript.

## Competing interests
The authors declare no competing interests.
