## [Peer Review File · Nature Communications]

Immune subset-committed proliferating cells populate the human foetal intestine throughout the second trimester of gestationREVIEWER COMMENTS

Reviewer #1 (Remarks to the Author):

In their manuscript titled "Immune subset-committed proliferating cells populate the human fetal intestine throughout the second trimester" Gao et al. examine immune cells in the human fetal intestine between gestational weeks (GW) 14 and 22. Using spectral flow cytometry they find that starting at GW15-17 the proportions of adaptive lymphocytes increase and eventually become dominant. Using imaging mass cytometry (IMC) the authors identify the presence of lymphoid follicles in the fetal intestine that contain ILC, and myeloid cells along with B and T cells. Further, LF T cells and ILC are shown to express c-KIT, IL-7R, CD117, and CCR6. The authors also identify Ki67+ cells within immune cell subsets by flow and IMC within LF and in the LP and show that the cells can proliferate ex vivo. While the topic is timely and the reviewer appreciates the preciousness of the samples, the study is purely descriptive providing no mechanistical insights into the nature or functionality of the cells in question.

Major points

- Again, the reviewer appreciates the preciousness of the samples but the data set would profit from an additional reference point of either a pediatric or adult specimen. Elmentaite et al. might also provide a reference.
- Throughout the paper the authors present flow cytometric and ICM analysis, which is rather limited given the exploratory purpose of the study. By the end of the discussion a previously unmentioned scRNAseq data set is presented for a very limited set of genes (SFig. 9, no M&M description, no mention on the data availability). scRNAseq analysis could substantially back up the presented findings and provide more depth into the presented results throughout the manuscript and could provide potential additional insights. The authors could, for example, follow up on their hypothesis of extrathymic T cell development by looking into the expression of TCR rearrangement genes (RAG, TDT etc.).
- The authors lay a strong focus on the expression of Ki67 in their data and argue that this readout reflects local proliferations and points for example towards extrathymic T cell development. However, Ki67 is not a binary marker of cell cycle, rather its levels decline gradually. Thus, it cannot be excluded that the T cells with the highest expression of Ki67 might have proliferated elsewhere (thymus?) and newly arrived in the intestine. Similar argumenta apply for the TREC analysis.
- It is not clear to the reviewer, which part of the intestine was taken for which analysis and whether it was always the same segment used for age dependent comparision, as cell composition and LF density likely vary substantially between for example small intestine and colon as well as along the individual segments.

Minor points

- The cell isolation protocol appears lengthy (digest @37°C over night). What is the percentage of surviving cells? Especially tissue resident cells do not like to prevail outside of tissue context and are prone to cell death and thus might have been excluded from flow cytometric analysis.
- Given the identification of markers for LF cells it would als be interesting to subject LP T cells to a more careful analysis in terms of their phenotype.
- Neonatal T cells in murine models have been shown to develop into effector rather than memory cell upon activation apparently pursuing the strategy of a vigorous immediate response over memory. TCF1 has recently emerged as a marker for stem like memory T cells. It would be interesting to find out whether the fetal intestinal T cells also harbor TCF1+ memory population that would likely persist long term.
- Figure 6A: Are the indicated parent gates right? It looks like the second gate arises from the CD20+ cells but should be from the negs etc.?
- Figure 7: Can the findings be backed up by statistical analysis?

- STable 7: Can the findings be backed up by statistical analysis?
- SFigure 9: Description lacking in the M&M section/data availability

Reviewer #2 (Remarks to the Author):

Title: Immune subset committed proliferating cells populate the human fetal intestine throughout the second trimester.

Summary: In this comprehensive manuscript on a subject that lacks information and is extremely difficult to evaluate. The authors evaluated immune cell types and their proliferation stages in the intestine of human fetuses from gestational weeks 14 to 22. The idea behind this study is that the intestine is the most significant part and source of the immune cells in humans; however, their development has not been studied in detail yet, due to the limits of the resources. Also, the authors wanted to evaluate how much of the gut's immune system is pre-developed before birth or whether it develops later. Authors have predominantly used spectral flow cytometry and image mass cytometry to evaluate the immune responses and found that myeloid cells and ILCs are the earliest cell types found during week 14. Still, later lymphocytes enrich the intestines and form lymphoid follicles rich in T- and B cells. The manuscript is nicely written, and the discussion is well explained.

There are a few major and minor comments:

1. Line 92: Expand IMDM and write its company.
2. One of my concerns is that the authors have used a percoll gradient to isolate cells. Percoll gradient is suitable for isolating mononuclear cells, but we can lose certain myeloid cell types.
3. Please be consistent in using "spectrum" or "spectral" flow cytometry. Line 96: Please change spectrum to spectral.
4. Line 101-104: This section looks incomplete. Authors have mentioned that they have used per/fix to fix and permeabilize cells, but they have not mentioned further intracellular antibody incubation and next steps.
5. The authors have mainly focused on CD45+ cells. At the initial stages of immune development, it could be essential to see CD45- cells as well. For example, megakaryocytes.
6. Minor: Line 209: in "Indicated", please change "I " to "I"
7. Line 212-214, figure 2: Based on the plots, maybe myeloid cells are not decreasing as much, but they appear due to the influx of T and B cells; thus percentage is going down. It is essential to include the total cell count of each cell type to evaluate real increases or decreases in specific populations.
8. A significant proportion of CD4+ T cell clusters is missing at week 21 (Figure 2A). Did the authors look at the reason for that?
9. It is crucial to see the overall percentage of CD45 changes over the weeks. We would recommend having a plot with total CD45 cells plotted against weeks. This will give readers accurate information about how overall cells change over time.
10. Suggestion: all the figures, rather than writing "different weeks", authors can write "gestation weeks"
11. Figure 6 and supplementary figure 8 should be combined. Maybe Figure 6B can be moved to supplementary.
12. For the cytokine results, there is no cytokine expression reported. There are several reasons why sometimes after stimulation cytokines are not seen by flow cytometry. When looking at the protocol for cytokine evaluation, the authors have incubated the cells for 2 days and then added brefeldin for 5 hours. It is possible that all the cytokines are released by the cells by that time. If the authors collected the supernatant, they should evaluate cytokine presence. Otherwise, it should be discussed and alternative and future directions specified.

Response to Reviewer Comments

(Manuscript ID: NCOMMS-22-12208-T)

Response to Reviewer #1 Comments

Reviewer #1 (Remarks to the Author):

In their manuscript titled "Immune subset-committed proliferating cells populate the human fetal intestine throughout the second trimester" Gao et al. examine immune cells in the human fetal intestine between gestational weeks (GW) 14 and 22. Using spectral flow cytometry they find that starting at GW15-17 the proportions of adaptive lymphocytes increase and eventually become dominant. Using imaging mass cytometry (IMC) the authors identify the presence of lymphoid follicles in the fetal intestine that contain ILC, and myeloid cells along with B and T cells. Further, LF T cells and ILC are shown to express c-KIT, IL-7R, CD117, and CCR6. The authors also identify Ki67+ cells within immune cell subsets by flow and IMC within LF and in the LP and show that the cells can proliferate ex vivo. While the topic is timely and the reviewer appreciates the preciousness of the samples, the study is purely descriptive providing no mechanistical insights into the nature or functionality of the cells in question.

Major points

- Again, the reviewer appreciates the preciousness of the samples but the data set would profit from an additional reference point of either a pediatric or adult specimen. Elmentaite et al. might also provide a reference.

Response: We appreciate the thorough review and agree with the reviewer that a reference to pediatric and adult samples is a valuable addition. We now make reference to a previous study of Stras et al¹ who have directly compared human fetal and infantile samples (Line 351-353). In addition, we have now incorporated observations from Elmentaite et al² into the discussion (Line 367-376).

- Throughout the paper the authors present flow cytometric and ICM analysis, which is rather limited given the exploratory purpose of the study. By the end of the discussion a previously unmentioned scRNAseq data set is presented for a very limited set of genes (SFig. 9, no M&M description, no mention on the data availability). scRNAseq analysis could substantially back up the presented findings and provide more depth into the presented results throughout the manuscript and could provide potential additional insights. The authors could, for example, follow up on their hypothesis of extrathymic T cell development by looking into the expression of TCR rearrangement genes (RAG, TDT etc.).

Response: We thank the reviewer for this suggestion. The scRNAseq dataset of fetal CD4 T cells was generated and in part described in our previous paper³. The single-cell RNA-seq data is available via Gene Expression Omnibus accession code GSE122846, which is now mentioned in the section of "Data availability". (Line 514) In the current study, we re-analyzed the gene expression profile of Ki-67⁺CD4⁺ T cells to support the observations from single cell suspension data and incorporated this analysis, together with the TREC analysis, into a new main figure in the revised manuscript (Figure 3).

We would like to point out that in our manuscript we hypothesized that thymus derived T cells expand locally in the developing intestine. We had not yet explored the idea of extrathymic T cell development. To investigate this, we have now followed the suggestion of the reviewer and analyzed the expression of genes involved in TCR rearrangement (*RAG1*, *RAG2*, *DNTT* and *LIG4*) in the scRNAseq dataset of fetal intestinal CD4⁺ T cells. We found only a few cells expressing *RAG1* and *LIG4*, and no expression of *RAG2* and *DNTT* (Figure 3d). Importantly, there were no cells co-expressing *RAG1* and *RAG2*, which is a requirement for rearrangements to take place, arguing against substantial T cell development in the fetal intestine. Hence, the most likely scenario is that fetal T cells develop in the thymus and then

migrate to the periphery where they expand further. This expansion could be taking place at an intermediate organ, such as mesenteric lymph nodes, or upon arrival in the intestines. We have also included these observations in the Discussion section (lines 439-443). Please see also our response to the comment below.

Figure 3

Figure 3. T cell receptor excisions circle (TREC) and single cell RNA-seq analysis confirms peripheral expansion of intestinal T cells. (a) Proliferative history in human fetal intestinal samples. (b-d) t-SNE embedding showing 1804 CD4⁺ T cells from a human fetal intestine analyzed by single-cell RNA-sequencing. The log-transformed expression levels of indicated immune markers (b), the phase of the cell-division cycle (c) and genes involved in TCR rearrangement (d) are shown. Each dot represents a single cell. The proliferative CD4⁺ T cells are encircled in (b).

- The authors lay a strong focus on the expression of Ki67 in their data and argue that this readout reflects local proliferations and points for example towards extrathymic T cell development. However, Ki67 is not a binary marker of cell cycle, rather its levels decline gradually. Thus, it cannot be excluded that the T cells with the highest expression of Ki67 might have proliferated elsewhere (thymus?) and newly arrived in the intestine. Similar argumenta apply for the TREC analysis.

Response: We agree with the reviewer on the interpretation of the data related to Ki-67 expression. While Ki-67 has been used extensively as a marker for proliferation in human tumors, its expression is variable throughout the cell cycle. Maximum levels are attained during the S-M phases, followed by a

decrease due to degradation in G1 and G0 (quiescence)⁴. Our data indicates that a substantial number of fetal intestinal immune cells express Ki-67 to levels that are detectable by flow cytometry and imaging mass cytometry (IMC). Given that the sensitivity of IMC is lower than that of flow cytometry, detection by IMC points to moderate to high levels of Ki-67 compatible with either active proliferation or recent entry into G1.

On the other hand, TREC analysis of fetal samples around 20-22 gestational weeks placed the replication history of T cells between 3 and 7 divisions (Figure 3a in the revised manuscript). Since the analysis performed in our study was designed to measure extrathymic proliferation, together with the absence of expression of TCR rearrangement genes (see previous point), we ascribed these divisions to extrathymic proliferation in the periphery.

Based on both Ki-67 expression and division history, we can envision two scenarios. In the first scenario, differentiated immune cells - including T cells- travel from the primary lymphoid organs to secondary lymphoid organs or intermediate tissues, where they undergo a few rounds of proliferation and then travel to the intestine. Given that Ki-67 is detectable at stable levels, the intestine would receive recently divided cells at a regular rate throughout the second trimester. In the second scenario, differentiated immune cells would reach the intestine at a regular rate during the second trimester, where they would receive local homeostatic proliferation signals and undergo a few rounds of division. In favor of the latter hypothesis, we have detected local IL-7 production in the fetal intestine, both in the epithelium and the lamina propria (data not shown). Additionally, the analysis of expression of cell cycle related genes in the previously generated scRNAseq dataset³ positions a significant number of fetal intestinal CD4⁺ T cells in the S or G2/M phase of the cell cycle (Figure 3c in the revised manuscript), suggesting they are actively proliferating in the tissue. We have now updated the Discussion section to include these ideas (Lines 450-459).

- It is not clear to the reviewer, which part of the intestine was taken for which analysis and whether it was always the same segment used for age dependent comparison, as cell composition and LF density likely vary substantially between for example small intestine and colon as well as along the individual segments.

Response: We apologize for this unclear description. In our studies, we have not aimed for analysis of particular intestinal segments, rather we have collected both the small intestine and colon from all samples included in the study. We have now mentioned this in the revised version of manuscript (Line 80-83).

Minor points

- The cell isolation protocol appears lengthy (digest @37°C over night). What is the percentage of surviving cells? Especially tissue resident cells do not like to prevail outside of tissue context and are prone to cell death and thus might have been excluded from flow cytometric analysis.

Response: We agree with the reviewer that extended cell isolation procedures can result in unwanted cell death, up to 10%. On the other hand, with shorter procedures there is the risk of suboptimal recovery of tissue resident cells. For this reason we previously optimized enzymatic digestion methods for liberating immune cells from tissues, based on the yield of live cells and conservation of the expression of CD markers for phenotyping. This has yielded the current protocol that has been successfully used in several previous publications from us^{3, 5, 6} while a similar protocol has also been used by others¹. Also, the composition of the immune system in different samples from the same gestational age is highly similar and we observed that the results of the single cell and imaging mass cytometry analysis are in line with each other. Therefore, we are confident that we do not lose certain cell subsets selectively.

- Given the identification of markers for LF cells it would be interesting to subject LP T cells to a more careful analysis in terms of their phenotype.

Response: Thanks for pointing that out. Indeed, we observed abundant expression of CD161, CD69, CD117 and CD127 in the LF (Figure R1). We fully agree with the reviewer that it is very interesting to further define the phenotype and function of these T cells within the LFs. We hope to report on these studies in a future publication.

Figure R1. The expression of CD161, CD69, CD117 and CD127 in human fetal intestinal samples by IMC. The LFs are indicated by the boxes.

- Neonatal T cells in murine models have been shown to develop into effector rather than memory cell upon activation apparently pursuing the strategy of a vigorous immediate response over memory. TCF1 has recently emerged as a marker for stem like memory T cells. It would be interesting to find out whether the fetal intestinal T cells also harbor TCF1+ memory population that would likely persist long term.

Response: The reviewer raises an interesting issue and we have now investigated the expression of TCF1 and the potential presence of stem cell-like memory cells (Tscm) in the fetal intestine. We

determined the expression of TCF1 in T cells from a week 20 gestational age fetal intestinal sample and compared it to TCF1 expression in adult PBMCs. We used an antibody panel which included CD45RA, CD45RO, CCR7 and CD95, to be able to identify naïve, stem-cell memory, central memory and effector/effector memory T cells, as depicted in **Figure R2**. As observed before, the fetal sample contained a low proportion of CD45RA⁺CCR7⁺ cells, compared to PBMCs. However, we were able to detect CD95⁺ cells within this subset, which corresponds to the Tscm phenotype in adult PBMCs⁷. TCF1 was expressed in all fetal T cell subsets. These results are compatible with the presence of stem cell memory within the fetal intestinal T cell compartment. The relatively high levels of TCF1 in all memory populations, including the Tem might be related to their highly proliferative potential and possibly recent egress from the thymus, as thymocytes express high levels of TCF1/TCF7.

Figure R2. TCF1 expression in naïve and memory subsets from fetal and adult origin.

• Figure 6A: Are the indicated parent gates right? It looks like the second gate arises from the CD20+ cells but should be from the negs etc.?

Response: We apologize for this mistake. Indeed, the second gate should be from the non B cell population. We have now corrected this in the revised Figure 7a.

• Figure 7: Can the findings be backed up by statistical analysis?

Response: Maybe the reviewer means Figure 6e? (Figure 7 in the revised manuscript) Indeed, these were analyzed by Wilcoxon signed-rank test for comparisons while no significant difference was found.

• Table 7: Can the findings be backed up by statistical analysis?

Response : Thanks for pointing it out. We have now added the standard deviation from the Q-PCR of $\delta\text{REC-}\psi\text{J}_\alpha$ in the Figure 3a of the revised manuscript.

- SFigure 9: Description lacking in the M&M section/data availability

Response : Thank you for pointing that out. We have now adjusted the section “Data availability”: “Single-cell RNA-seq data is available via Gene Expression Omnibus accession code GSE122846. The flow cytometry data are available via Flow Repository (<http://flowrepository.org/id/FR-FCM-Z5MB>). Imaging mass cytometry data are deposited at Mendeley Data (<https://data.mendeley.com/v1/datasets/m4vr79wsjs/draft?preview=1>).” in the revised manuscript. (line 514-517)

Response to Reviewer #2 Comments

Reviewer #2 (Remarks to the Author):

Title: Immune subset committed proliferating cells populate the human fetal intestine throughout the second trimester.

Summary: In this comprehensive manuscript on a subject that lacks information and is extremely difficult to evaluate. The authors evaluated immune cell types and their proliferation stages in the intestine of human fetuses from gestational weeks 14 to 22. The idea behind this study is that the intestine is the most significant part and source of the immune cells in humans; however, their development has not been studied in detail yet, due to the limits of the resources. Also, the authors wanted to evaluate how much of the gut's immune system is pre-developed before birth or whether it develops later. Authors have predominantly used spectral flow cytometry and image mass cytometry to evaluate the immune responses and found that myeloid cells and ILCs are the earliest cell types found during week 14. Still, later lymphocytes enrich the intestines and form lymphoid follicles rich in T- and B cells. The manuscript is nicely written, and the discussion is well explained.

There are a few major and minor comments:

1. Line 92: Expand IMDM and write its company.

Response 1: As suggested, we have modified "After incubation, the cell preparations were filtered through a 70 μm cell strainer (Corning) followed by washing of the cells with Iscove's Modified Dulbecco's Medium (IMDM, Lonza)." in the revised manuscript. (Line 89-91)

2. One of my concerns is that the authors have used a percoll gradient to isolate cells. Percoll gradient is suitable for isolating mononuclear cells, but we can lose certain myeloid cell types.

Response 2: We agree with the reviewer that this is a possibility. However, we used a protocol that has been successfully used in several previous publications from us^{3,5,6} while a similar protocol has also been used by others¹. Here, percoll is used as this can separate the immune cells from the highly abundant non immune cells and debris which would otherwise potentially interfere with the single cell analysis. Using this method, we could identify a myeloid cell population but the antibody panel was not designed to capture the heterogeneity in the myeloid compartment. However, we were able to visualize several myeloid cell populations with imaging mass cytometry. We anticipate that in future work we will be able to shed more light on the heterogeneity of the myeloid cells in the developing fetal intestine.

3. Please be consistent in using "spectrum" or "spectral" flow cytometry. Line 96: Please change spectrum to spectral.

Response 3: Based on the reviewer's suggestion, we have changed spectrum to spectral. (Line 95)

4. Line 101-104: This section looks incomplete. Authors have mentioned that they have used per/fix to fix and permeabilize cells, but they have not mentioned further intracellular antibody incubation and next steps.

Response 4: We apologize for this omission. We have now added "For intracellular staining, the fixed/permeabilized cells were incubated with the antibodies for 45 min at 4 °C, followed by washing of the cells with permeabilization buffer. Then the stained cells were resuspended." in the revised version of manuscript (line 103-105).

5. The authors have mainly focused on CD45⁺ cells. At the initial stages of immune development, it could be essential to see CD45⁻ cells as well. For example, megakaryocytes.

Response 5: We fully agree with the reviewer that the CD45⁻ cells is an important population to be studied, however, due to the lack of the specific markers of CD45⁻ cells in the flow cytometry and

imaging mass cytometry antibody panels our current analysis did not focus on this cell subset. This will be the subject of future studies.

6. Minor: Line 209: in “Indicated”, please change “l “ to “l”

Response 6: Thanks for pointing out. We have changed “Indicated” to “indicated” in the revised manuscript. (Line 203).

7. Line 212-214, figure 2: Based on the plots, maybe myeloid cells are not decreasing as much, but they appear due to the influx of T and B cells; thus percentage is going down. It is essential to include the total cell count of each cell type to evaluate real increases or decreases in specific populations.

Response 7: As suggested, we have now added the graphs of the total cell count of each cell type as Supplementary Fig. 4 in the revised manuscript.

Supplementary Figure. 4 Overview of the total cell count per indicated immune lineage from all human fetal intestine samples analyzed (n = 28).

8. A significant proportion of CD4+ T cell clusters is missing at week 21 (Figure 2A). Did the authors look at the reason for that?

Response 8: Indeed, we have also noted that the sample from week 21 in Figure 2A deviates from the other samples in that figure and also from the samples analyzed in the 2 additional experiments. This is probably not related to that gestational week, since the two additional intestinal samples from week 21 included in the study (Supplementary Figure 3a, b) have a CD4 T cell compartment that looks very comparable to the other samples. Unfortunately, due to privacy issues, we have no information on the source of the material other than that it is from elective abortions. Developmental and chromosomal abnormalities could lead to an elective abortion and could underlie such differences. All in all, we believe that this particular sample is not representative. However, we did not choose to omit it from the analysis as we felt this would not be appropriate.

9. It is crucial to see the overall percentage of CD45 changes over the weeks. We would recommend having a plot with total CD45 cells plotted against weeks. This will give readers accurate information about how overall cells change over time.

Response 9: As suggested, we have now added the graphs of the percentage of CD45+ immune cells within single live cells over the weeks as Fig. 2b in the revised manuscript.

Figure 2

Figure 2. Presence of Ki-67⁺ cells throughout gestational week 14 to 22.

(a) Display of the Ki-67 expression in the optSNE plots of the individual fetal intestinal samples from week 14 through 22. Colors represent relative expression of Ki-67. Data are representative of three

independent experiments. (b) The frequency of the CD45⁺ immune cells within single alive cells of all human fetal intestine samples analyzed (n = 28). (c-i) Overview of the frequency of each immune lineage within the CD45⁺ immune cells of all human fetal intestine samples analyzed (n = 28). (j) Overview of the distribution of Ki-67⁺ cells in the indicated immune lineages from gestational week 14 through 22, the results shown are from the samples shown in panel (a). (k) The percentage of Ki-67⁺ cells within the CD45⁺ immune cells of all human fetal intestine samples analyzed (n = 28).

10. Suggestion: all the figures, rather than writing “different weeks”, authors can write “gestation weeks”

Response 10: Thank you for the comments. We have now changed “ different weeks” into “gestation weeks” in all Figures and Supplementary Figures of the revised manuscript.

11. Figure 6 and supplementary figure 8 should be combined. Maybe Figure 6B can be moved to supplementary.

Response 11: As suggested, we have now modified Figure 7 (previously Figure 6) and Supplementary Figure 9 (previously supplementary Fig. 8) in the revised manuscript.

Figure 7

Figure 7. Fetal intestinal samples harbor cells with proliferative capacity in vitro

(a) Biaxial plots showing the gating strategy to identify immune subsets in a human fetal intestinal sample from gestational week 16 cultured in medium alone or medium with IL-7 for 4 days. The

colored gates indicate the identified immune subsets. Data represent four independent experiments. (b) Histogram showing CellTrace™ Violet dye dilution in the indicated immune subsets from the human fetal intestine. Dashed lines represent the fetal intestinal sample cultured in medium, while solid lines represents the fetal intestinal sample cultured in medium with IL-7. (c) Histogram showing CellTrace™ Violet dye dilution by the indicated immune subsets from a PBMCs sample. Dashed lines represent the PBMCs cultured in medium, while solid lines represents the PBMCs cultured in medium with IL-7. (d) Cell counts of the immune subset indicated from human fetal intestinal samples in both culture conditions. Data were from four independent experiments. (e) Comparison of the mean MFI values of CD40L expression between culture in medium alone and in the presence of IL-7. (f) Comparison of the percentages of CD40L⁺granzyme B⁺ expressing cells within CD3⁺CD7⁺ ILCs, CD4⁺CD8⁻ T cells, CD4⁺ T cells and CD8⁺ cells in both culture conditions. Error bars indicate mean ± s.e.m. *P < 0.05, Wilcoxon signed-rank test for comparisons.

Supplementary Figure 9 Functional profiling of fetal intestinal cells

(a-d) Fetal intestinal cells were cultured in the absence or presence IL-7 for 48 hours followed by flow cytometric analysis of the expression of CD40L, granzyme B, TNF- α , IFN γ , IL-2, and IL-17A in CD3⁻CD7⁺ ILCs and T cells (CD4⁻CD8⁻ T, CD4⁺ T and CD8⁺ T). In total, four independent experiments were performed. (e) Intracellular expression of granzyme B and CD40L was determined for CD3⁻CD7⁺ ILCs and T cells (CD4⁻CD8⁻ T, CD4⁺ T and CD8⁺ T) by flow cytometry in both culture conditions. The biaxial plots show data from one representative experiment.

12. For the cytokine results, there is no cytokine expression reported. There are several reasons why sometimes after stimulation cytokines are not seen by flow cytometry. When looking at the protocol for cytokine evaluation, the authors have incubated the cells for 2 days and then added brefeldin for 5 hours. It is possible that all the cytokines are released by the cells by that time. If the authors collected the supernatant, they should evaluate cytokine presence. Otherwise, it should be discussed and alternative and future directions specified.

Response 12: Thank you for pointing it out. Unfortunately, we did not collect the supernatant from these experiments. The purpose of the experiments was to investigate whether the fetal intestinal cells would show other signs of activation, while proliferating in the absence or presence of IL-7, as it happens when they are stimulated via the TCR³. We did several experiments at day 2, where we saw extensive proliferation in the condition with IL-7. At this time point, we saw clear increases in CD40L and granzyme B expression. However, the proliferating cells did not produce the evaluated cytokines. We agree with the reviewer that cytokines might already have been released by then. In response to the question of the reviewer we have now evaluated cytokine production after 6 and 24 hours culture in the presence IL-7. Again, we did not detect cytokine production, while CD40L and granzyme B expression were already detectable at 24 hours. Our results indicate that proliferation is ongoing without secretion of pro- or anti-inflammatory cytokines, which could underlie maintenance of homeostasis in the developing fetal intestine. We have now added a sentence to clarify this in the discussion. Line (468-470)

Reference

1. Stras, S.F. *et al.* Maturation of the Human Intestinal Immune System Occurs Early in Fetal Development. *Dev Cell* **51**, 357-373.e355 (2019).
2. Elmentaite, R. *et al.* Cells of the human intestinal tract mapped across space and time. *Nature* **597**, 250-255 (2021).
3. Li, N. *et al.* Memory CD4(+) T cells are generated in the human fetal intestine. *Nat Immunol* **20**, 301-312 (2019).
4. Miller, I. *et al.* Ki67 is a Graded Rather than a Binary Marker of Proliferation versus Quiescence. *Cell Rep* **24**, 1105-1112.e1105 (2018).
5. Li, N. *et al.* Early-Life Compartmentalization of Immune Cells in Human Fetal Tissues Revealed by High-Dimensional Mass Cytometry. *Front Immunol* **10**, 1932 (2019).
6. Li, N. *et al.* Mass cytometry reveals innate lymphoid cell differentiation pathways in the human fetal intestine. *J Exp Med* **215**, 1383-1396 (2018).

7. Gattinoni, L., Speiser, D.E., Lichterfeld, M. & Bonini, C. T memory stem cells in health and disease. *Nat Med* **23**, 18-27 (2017).

REVIEWER COMMENTS

Reviewer #1 (Remarks to the Author):

In the revised version of their manuscript "Immune subset-committed proliferating cells populate the human fetal intestine throughout the second trimester" Guo and colleagues examine immune cells isolated from the fetal intestine and describe developmental kinetics of innate and adaptive immune cells between gestational weeks 16 and 22. Further, they report that each population of immune cells also consists of a subpopulation that expresses the proliferation associated protein Ki67 and the authors identify a subpopulation of innate and adaptive cells with common markers that are organized in gut associated lymphoid structures of undefined nature. Also, the authors show that unlike adult PBMC counterparts fetal cells spontaneously proliferate ex vivo. The study is highly descriptive and provides little mechanistic insight, which in part is of course due to the nature of the specimen. However, as previously mentioned the dataset would profit from the inclusion of an adult or pediatric specimen to link the developmental differences or similarities to the situation ex utero. The authors spend a lot of resources to show that there is Ki67 expression among all the subsets of cells identified. This is not unexpected, even in the adult situation as Ki67 expression gradually decreases after cell division thus the detection of its expression does not provide a better understanding of the location or the reason of cell division detected. The authors report that a good proportion of the cells within their single cell suspension of the fetal intestine are organized in GALT structures. The nature of the GALT structures is not defined further. Are those multifollicular structures that rather resemble Peyer's patches of the SI or colonic patches or are those rather single follicles or any of those. The authors do not even discriminate between small intestinal and colonic specimen within their data set. There are arguments that can be raised that during fetal development these two tissues are more similar but in principle these are different organs with different functions, microbial exposure and definitely different immune cell composition ex utero.

Minor points:

Figure 7a: The indicated parent gates are (still) not correct (e.g. T cells arise from CD3- etc.)
Figures 7b vs. d: Is the effect of IL7 on proliferation now significant or not? In Figure 7b it appears like there are differences but in 7d the statistics argue against that.

Reviewer #3 (Remarks to the Author):

This is a much improved version of the original submission, competently answering the majority of the main comments, and enriched with additional quality data and discussion. The fact remains that this is a purely observational study, albeit at a high level, given the quality of the data presented and the rare nature of the samples involved.

Response to Reviewers' Comments (Manuscript ID: NCOMMS-22-12208A)

Response to Reviewer #1 Comments

Reviewer #1 (Remarks to the Author):

In the revised version of their manuscript “Immune subset-committed proliferating cells populate the human fetal intestine throughout the second trimester” Guo and colleagues examine immune cells isolated from the fetal intestine and describe developmental kinetics of innate and adaptive immune cells between gestational weeks 16 and 22. Further, they report that each population of immune cells also consists of a subpopulation that expresses the proliferation associated protein Ki67 and the authors identify a subpopulation of innate and adaptive cells with common markers that are organized in gut associated lymphoid structures of undefined nature. Also, the authors show that unlike adult PBMC counterparts fetal cells spontaneously proliferate *ex vivo*. The study is highly descriptive and provides little mechanistic insight, which in part is of course due to the nature of the specimen.

However, as previously mentioned the dataset would profit from the inclusion of an adult or pediatric specimen to link the developmental differences or similarities to the situation *ex utero*.

Response: As the reviewer suggested, we have collected fresh unaffected biopsy specimens from ileum and colon of two pediatric and three adult patients. Single-cell suspensions from these biopsies were stained and analyzed by spectral flow cytometry. tSNE was used to identify the major immune subsets and visualize selected marker expression profiles (**Figure R1a-c**). Quantification of the composition of the immune compartment in the fetal, pediatric and adult samples shows that the frequency of CD3⁻CD7⁺ ILCs is higher in early gestational weeks, to then decrease to the levels found in pediatric and adult samples after week 16 (**Figure R1d-f**). After gestational week 16, over 50% of immune cells were CD4⁺ T cells whereas CD8⁺ T cells comprised around 10-15% of the total in human fetal intestines (**Figure R1d-f**). In contrast, in both the pediatric and adult samples we observed a lower proportion of CD4⁺ T cells, while frequencies of CD8⁺ T cells were increased (**Figure R1d-f**).

Within the lymphoid compartment, based on expression of CCR7, CD45RA and CD45RO, we observed naïve and memory T cells in both pediatric and adult samples (**Figure R1a-c**). In addition, comparison of expression of CD127/IL-7R α on the major lymphoid immune subsets in the fetal, pediatric and adult samples revealed higher expression in the fetal samples (**Figure R1g**). Thus, clear differences exist in the abundance of subsets and the expression of cell surface molecules on these subsets between the fetal and *ex-utero* samples. We have now added these data as Supplementary Fig. 5 in the revised manuscript.

Figure R1. Comparison of fetal, pediatric and adult intestinal samples by spectral flow cytometry. (a-c) tSNE plots of the data from (a) fetal, (b) pediatric and (c) adult samples, where the major immune subsets are indicated and the expression of selected markers is shown. (d-f) Percentages of the major immune subsets in the samples analyzed. (g) Frequency of CD127/IL-7R α expressing cells within CD3⁺CD7⁺ ILCs, CD4⁺CD8⁻ DN T cells, CD4⁺ T cells and CD8⁺ T cells in fetal, pediatric and adult samples. Each dot represents an individual sample. Error bars indicate mean \pm SD.

The authors spend a lot of resources to show that there is Ki67 expression among all the subsets of cells identified. This is not unexpected, even in the adult situation as Ki67 expression gradually decreases after cell division thus the detection of its expression does not provide a better understanding of the location or the reason of cell division detected.

Response: We understand the concern of the reviewer regarding Ki-67 expression. We have indeed pointed out that Ki-67 is expressed at increasing levels during the cell cycle, peaking at the S and G2/M phase, to gradually decrease thereafter (lines 466-468). Detection of Ki-67 expression thus suggests that the cells are not quiescent. When we analyzed the expression of cell cycle-related genes in the previously generated scRNAseq dataset, a significant number of fetal intestinal CD4⁺ T cells were assigned to S or G2/M phase of the cell cycle (**Figure 3c**), suggesting they are actively proliferating in the tissue. Comparison of Ki-67 expression on immune cells from fetal, pediatric and adult samples revealed that, for each major immune subset, we are able to observe multiple Ki-67⁺ pockets in human fetal intestinal samples (**Figure R2a**), while less were detected in pediatric and adult samples (**Figure R2b-c**). In agreement, the frequency of Ki-67⁺ immune cells was lower in pediatric and adult samples compared to fetal samples (**Figure R2d**), suggesting that proliferation might be occurring at a higher rate in the fetal intestine.

With regards to location, we have shown that Ki-67⁺ cells are located in the lamina propria and in lymphoid structures (**Figure 6a-d**). Also, we observed that a higher percentage of lymphoid cells express CD127/IL-7R α compared to the pediatric and adult samples (**Figure R1g**). Moreover, with RNAscope analysis we now provide evidence for the presence of IL-7 mRNA both in the fetal lamina propria and epithelium (**Figure 6e-g**), and show that, unlike adult circulatory cells, fetal cells proliferate vigorously in response to IL-7. Altogether, this supports a scenario similar to lymphopenia-induced proliferation where different fetal lymphoid subsets would be exposed to IL-7 upon arrival in the intestine and proliferate to fill up the organ as it grows.

Figure R2. Presence of Ki-67⁺ immune cells in fetal, pediatric and adult samples. (a-c) tSNE plots displaying Ki-67 expression in samples from (a) fetal, (b) pediatric and (c) adult samples. (d) Percentage of Ki-67⁺ cells within CD45⁺ immune cells of all samples analyzed.

The authors report that a good proportion of the cells within their single cell suspension of the fetal intestine are organized in GALT structures. The nature of the GALT structures is not defined further. Are those multi-follicular structures that rather resemble Peyer's patches of the SI or colonic patches or are those rather single follicles or any of those.

Response: In response to the question of the reviewer, we have now highlighted the presence of two isolated lymphoid follicles (CD45 and CD20 expression, **Figure R3a**) in a fetal intestinal sample from gestational week 17, shown in **Figure 4**. We have also performed RNAscope assays on a human fetal intestine from gestational week 14 to detect CXCL13 expression. CXCL13 has been shown to be expressed by lymphoid tissue organizer-like stromal cells in the fetal gut¹. We observed abundant CXCL13 transcripts within round structures located below the epithelium, similar to the single follicles detected by IMC (**Figure R3b**, also included in the revised manuscript as **Figure 4g**). Thus, these data point to scattered isolated lymphoid follicles in human fetal intestines. According to an early study from 1965, fetal human ileum has been shown to contain Peyer's patches with more than 5 follicles at gestational week 24². Thus far, we have not encountered multi-follicular structures in the samples analyzed in this work, ranging from gestational week 14 to 22. Formation of multi-follicular structures is thus likely to occur after week 22.

Figure R3. Identification of isolated single lymphoid follicles on human fetal intestine by IMC and RNAscope. **(a)** Visualization of lymphoid follicles (indicated by boxes) of the fetal intestinal sample from week 17 by CD45 and CD20 staining. Scale bar, 200 μm . **(b)** Detection of CXCL13 transcripts by RNAscope in a human fetal intestine from week 14.

The authors do not even discriminate between small intestinal and colonic specimen within their data set. There are arguments that can be raised that during fetal development these two tissues are more similar but in principle these are different organs with different functions, microbial exposure and definitely different immune cell composition ex utero.

Response: As pointed out by the reviewer, the composition of the intestinal immune compartment has been described to differ markedly between intestinal segments. While most of the studies have been performed in mice, recent work on human endoscopic material from different colonic segments in healthy volunteers also indicated differences³. While in the absence of significant microbial exposure such differences may be less pronounced in the developing fetus we agree with the reviewer that additional information of prenatal immune cell subset compartmentalization in intestinal segments would be interesting. However, at the time we were collecting the fetal samples our primary aim was to obtain information on the development of the immune compartment throughout the second trimester, and fetal intestinal samples were processed without separation of proximal, central or distal segments. At present we have no additional material available but we will attempt to address this important issue in future studies.

Minor points:

Figure 7a: The indicated parent gates are (still) not correct (e.g. T cells arise from CD3- etc.)

Response: We thank the reviewer for pointing out these errors. We have now modified Figure 7a in the revised manuscript.

Figure 7. Fetal intestinal samples harbor cells with proliferative capacity in vitro. (a) Biaxial plots showing the gating strategy to identify immune subsets in a human fetal intestinal sample from gestational week 16 cultured in medium alone or medium with IL-7 for 4 days. The colored gates indicate the identified immune subsets. Data represent four independent experiments.

Figures 7b vs. d: Is the effect of IL7 on proliferation now significant or not? In Figure 7b it appears like there are differences but in 7d the statistics argue against that.

Response: We agree this was confusing and therefore consulted a biostatistician in our institute who advised to use the Mann-Whitney test as all the values in the IL-7 condition are above the values in the control condition. As Figure 7b shows CTV dilution, we have now aligned Fig 7b and 7d and quantified the percentage of CTV^{diluted} cells in the major immune subsets (Figure R4, also as Figure 7d in the revised manuscript) and used the Mann-Whitney test which revealed significant differences in CD127/IL-7R positive lymphoid subsets.

Figure R4. Quantification of CTV^{diluted} cells for the major immune subsets after culture in medium alone or in medium with IL-7. * P < 0.05, Mann-Whitney test for comparisons.

Response to Reviewer #3 Comments

Reviewer #3 (Remarks to the Author):

This is a much improved version of the original submission, competently answering the majority of the main comments, and enriched with additional quality data and discussion. The fact remains that this is a purely observational study, albeit at a high level, given the quality of the data presented and the rare nature of the samples involved.

Response: We appreciate these positive comments from the reviewer.

1. Elmentaite, R. *et al.* Cells of the human intestinal tract mapped across space and time. *Nature* **597**, 250-255 (2021).
2. Cornes, J.S. Number, size, and distribution of Peyer's patches in the human small intestine: Part I The development of Peyer's patches. *Gut* **6**, 225-229 (1965).
3. Tyler, C.J. *et al.* Inherent Immune Cell Variation Within Colonic Segments Presents Challenges for Clinical Trial Design. *J Crohns Colitis* **14**, 1364-1377 (2020).

REVIEWERS' COMMENTS

Reviewer #1 (Remarks to the Author):

In their revised version of the manuscript titled "Immune subset-committed proliferating cells populate the human fetal intestine throughout the second trimester" Guo and colleagues characterize the hematopoietic cells in the lamina propria of a not closer defined regions of the fetal intestine between gestational week 14 to 22 and document the dynamics of immune cells during that time. Further, they note that a significant proportion of innate and adaptive immune cells are organized in lymphoid follicles and are able to phenotype innate lymphocytes and T cells within the follicles expressing a certain subset of markers (i.e. CD69, CD161, CD117, CCR6 and CD127) that separates them from the LP counterparts. Also, they report that various lymphocyte populations of the fetal intestine but not adult PBMCs proliferate in response to IL-7. The findings might contribute to our understanding of intestinal immune development. The criticism of the reviewer, however, persists that data are somewhat disconnected from the ex utero development and the strong focus on the analysis of Ki67 deflects the focus from the, in the reviewer's opinion, more important findings that are found in the second half of the manuscript. The authors have addressed the first part of my criticism and added data on immune cell composition from pediatric and adult samples but those are hidden far in the supplements. They would contribute to a more global understanding in immune cell changes when shown more prominently. The same applies for the Ki67, the expression of Ki67 per se is not interpretable but the comparison to the pediatric/adult samples is more conclusive, but also here the comparison is to be found in the supplement.

Minor points

Fig. 7: In the rebuttal the authors say that they used the Mann Whitney U test, in the figure legend Wilcoxon signed-rank test is indicated. The latter would be the correct one to use for paired comparison.

Fig. S4: Are the absolute numbers somehow normalized for example to the weight/area of the specimen? This is not stated and if not normalized they cannot be compared and should be omitted.

Response to Reviewers' Comments

(Manuscript ID: NCOMMS-22-12208B)

Reviewer #1 (Remarks to the Author):

In their revised version of the manuscript titled "Immune subset-committed proliferating cells populate the human fetal intestine throughout the second trimester" Guo and colleagues characterize the hematopoietic cells in the lamina propria of a not closer defined regions of the fetal intestine between gestational week 14 to 22 and document the dynamics of immune cells during that time. Further, they note that a significant proportion of innate and adaptive immune cells are organized in lymphoid follicles and are able to phenotype innate lymphocytes and T cells within the follicles expressing a certain subset of markers (i.e. CD69, CD161, CD117, CCR6 and CD127) that separates them from the LP counterparts. Also, they report that various lymphocyte populations of the fetal intestine but not adult PBMCs proliferate in response to IL-7. The findings might contribute to our understanding of intestinal immune development. The criticism of the reviewer, however, persists that data are somewhat disconnected from the ex utero development and the strong focus on the analysis of Ki67 deflects the focus from the, in the reviewer's opinion, more important findings that are found in the second half of the manuscript. The authors have addressed the first part of my criticism and added data on immune cell composition from pediatric and adult samples but those are hidden far in the supplements. They would contribute to a more global understanding in immune cell changes when shown more prominently. The same applies for the Ki67, the expression of Ki67 per se is not interpretable but the comparison to the pediatric/adult samples is more conclusive, but also here the comparison is to be found in the supplementary.

Response: Thanks for reviewer's comments. As the reviewer suggested, we have now moved Supplementary Fig. 5 to Fig. 3 to show the comparison of the immune cell composition from fetal, pediatric and adult samples.

Minor points

Fig. 7: In the rebuttal the authors say that they used the Mann Whitney U test, in the figure legend Wilcoxon signed-rank test is indicated. The latter would be the correct one to use for paired comparison.

Response: We thank the reviewer for pointing it out. We used Wilcoxon signed-rank test for the comparison.

Fig. S4: Are the absolute numbers somehow normalized for example to the weight/area of the specimen? This is not stated and if not normalized they cannot be compared and should be omitted.

Response: As the reviewer suggested, we have removed Supplementary Fig. 4.